# CROSS-DOMAIN REINFORCEMENT LEARNING VIA PREFERENCE CONSISTENCY

## ABSTRACT

Cross-domain reinforcement learning (CDRL) aims to utilize the knowledge acquired from a source domain to efficiently learn tasks in a target domain. Unsupervised CDRL assumes no access to any signal (*e.g.*, rewards) from the target domain, and most methods utilize state-action correspondence or cycle consistency. In this work, we identify the critical correspondence identifiability issue (CII) that arises in existing unsupervised CDRL methods. To address this identifiability issue, we propose leveraging pairwise trajectory preferences in the target domain as weak supervision. Specifically, we introduce the principle of *cross-domain preference consistency* (CDPC)–a policy is more transferable across the domains if the source and target domains have similar preferences over trajectories–to provide additional guidance for establishing proper correspondence between the source and target domains. To substantiate the principle of CDPC, we present an algorithm that integrates a state decoder learned through preference consistency loss during training with a cross-domain MPC method for action selection during inference. Through extensive experiments in both MuJoCo and Robosuite, we demonstrate that CDPC enables effective and data-efficient knowledge transfer across domains, outperforming state-of-the-art CDRL benchmark methods.

## 1 INTRODUCTION

Reinforcement Learning (RL) has shown impressive success on a wide range of tasks, encompassing both discrete and continuous control scenarios, such as game playing (Mnih et al., 2015; Silver et al., 2016; Vinyals et al., 2019) and robot control (Levine et al., 2016; Tobin et al., 2017). However, solving these tasks in a data-efficient manner has remained a significant challenge in RL, mainly due to the need for extensive online trial-and-error interactions and the resulting prolonged training periods. To alleviate the data efficiency issue, one natural and promising approach is to reuse the control policies learned on similar tasks for fast knowledge transfer. Built on this intuition, cross-domain reinforcement learning (CDRL) offers a generic formulation that extends the applicability of transfer learning to RL, where the source domain and the target domain can have different transition dynamics or distinct state-action spaces. With access to the source domain (*e.g.*, the data samples or the environment) and the pre-trained source-domain models (*e.g.*, policies or value functions), CDRL aims to transfer the knowledge acquired from the source domain to improve the sample efficiency in the target domain. This adaptability of CDRL is crucial for overcoming the data inefficiency in conventional RL, offering a more flexible and resource-efficient solution.

Several attempts on CDRL (Zhang et al., 2021a; Gui et al., 2023) have demonstrated the possibility of direct policy transfer by learning the state-action correspondence between domains, or essentially inter-domain mapping functions, from unpaired trajectories in a fully unsupervised manner, *i.e.*, no reward signal available in the target domain. For example, (Zhang et al., 2021a) proposes to learn the state-action correspondence (*i.e.*, a target-to-source state decoder and a source-to-target action encoder) by minimizing a dynamics cycle consistency loss, which aligns the one-step transition of the unpaired trajectories from the two domains. These unsupervised approaches can serve as powerful RL solutions in practice as it is widely known that reward design can require substantial efforts and hence is rather time-consuming. However, we identify that this unsupervised approach can be prone to the *correspondence identifiability issue* (CII). This phenomenon indicates that without any supervision from the target domain, learning the state-action correspondence can be an underdetermined problem. To illustrate this, we provide a toy example of a gridworld as shown in Figure 1. Motivated by this,

we want to tackle this research question: *How to address the correspondence identifiability issue in cross-domain transfer for RL with only weak supervision?*

In this paper, we answer the above question from the perspective of *cross-domain preference-based RL* (CD-PbRL). Specifically, we present a new CDRL setting where the agent in the target domain can receive additional weak supervision signal in the form of *preferences over trajectory pairs*. In the context of RL, a weakly-supervised setting refers to scenarios where the learners rely on indirect supervision, such as human preferences or rankings, rather than explicit reward labels, to learn well-performing policies (Lee et al., 2020; Wang et al., 2022). Inspired by the classic preference-based RL (PbRL) (Wirth & Fürnkranz, 2013; Wirth et al., 2017) and the recent works on the fine-tuning of language models (Stiennon et al., 2020; Ouyang et al., 2022), we posit that preference feedback can serve as feasible surrogate supervision to tackle the identifiability issue in CDRL. Our insight is that pairwise preference implicitly encodes the underlying goal of the task, and hence the *consistency in preference* across the source and target domains indicates their domain similarity. Accordingly, we propose the framework of *Cross-Domain Preference Consistency* (CDPC), which

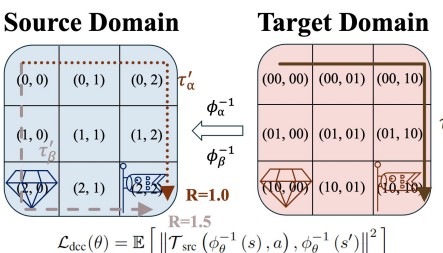

$$\mathcal{L}_{\text{dcc}}(\theta) = \mathbb{E}\left[\left\|\mathcal{T}_{\text{src}}\left(\phi_\theta^{-1}(s), a\right), \phi_\theta^{-1}(s')\right\|^2\right]$$

Figure 1: **An illustrative example of the correspondence identifiability issue:** In a $3 \times 3$ gridworld, the source domain (decimal) and target domain (binary) share the same structure: the start is the top-left, the treasure (+0.5) is on the bottom-left, and the goal (+1, ends the episode) is on the bottom-right. Two state decoders, $(\phi_\alpha^{-1})$ and $(\phi_\beta^{-1})$, map $\tau$ into $\tau'_\alpha$ and $\tau'_\beta$, both ensuring transitions via $\pi_{\text{src}}$ with zero dynamics cycle consistency loss since $\phi^{-1}(s_t)$ via $\pi_{\text{src}}$ matches $\phi^{-1}(s_{t+1})$ exactly. However, identifying the better decoder based only on dynamics cycle consistency loss appears infeasible, revealing an identifiability issue. The detailed explanation is provided in Appendix.

can better learn the state-action correspondence by enforcing the trajectory preferences to be aligned across the two domains, based on the intuition that a policy is transferable across domains if the source and target domains have better consensus on the preference over trajectories under some inter-domain mapping.

The proposed CDPC framework consists of two major components: (i) *Target-to-source state decoder*: To enable the reuse of a source-domain pre-trained policy (denoted by $\pi_{\text{src}}$), CDPC learns a target-to-source state decoder (denoted by $\phi^{-1}$). To learn $\phi^{-1}$ without suffering from CII, CDPC utilizes a cross-domain pairwise preference loss (or equivalently the negative log-likelihood), which is calculated with respect to the source-domain trajectories induced by $\phi^{-1}$ with the target-domain preferences as our labels. Compared to the existing unsupervised CDRL, this loss function offers additional constraints for the state decoder such that the identifiability issue can be mitigated. (ii) *Cross-domain model predictive control for inference*: During inference, we propose to leverage the learned state decoder and determine the target-domain actions by *planning* via model-predictive control (MPC). Specifically, at each time step, we generate multiple synthetic target-domain trajectories of finite length (with the help of a learned dynamics model) and choose the first action of the best trajectory. Different from the standard MPC, the proposed cross-domain MPC uses the *source-domain reward* of the source-domain trajectory induced by the state decoder as the selection criterion for MPC. With this design, there is no need to learn the action correspondence between source and target domains. Moreover, this framework is general, *i.e.*, that it can be integrated with any enhancements of MPC.

We evaluate CDPC against various CDRL benchmark methods on various tasks in MuJoCo and Robosuite. The main observations are: (1) Through preference consistency, CDPC achieves faster and more stable learning curves in training the state decoder than the other CDRL methods. (2) Additionally, CDPC enjoys superior sample efficiency across different dataset sizes, even when compared to the baselines with true reward information. (3) We also provide several ablation studies, confirming the significance of the preference consistency loss and examining the impact of the proportions of expert data on CDPC. (4) Moreover, we perform additional experiments to investigate the effect of the quality of preference labels on CDPC. Interestingly, by randomly perturbing a portion of the preference labels, we found that CDPC can still achieve reliable cross-domain transfer under

certain perturbation ratios. (5) Finally, we further corroborate the strong cross-domain transferability of CDPC through experiments under various domain similarities.

## 2 RELATED WORK

Cross-domain transfer in RL (Taylor & Stone, 2009; Zhu et al., 2023; Serrano et al., 2024; Lyu et al., 2024; Wen et al., 2024; Tian, Hongduan and Liu, Feng and Liu, Tongliang and Du, Bo and Cheung, Yiu-ming and Han, Bo, 2024) is an area of research within RL that specifically addresses the challenge of transferring learned policies or value functions from one domain to another, even when there are disparities in state-action dimensions between the domains. Cross-domain transfer learning can be divided into imitation learning (Kim et al., 2020; Fickinger et al., 2021; Raychaudhuri et al., 2021) and transfer learning. Transfer learning itself can be further categorized into single-source transfer (Ammar & Taylor, 2012) and multiple-source transfer (Ammar et al., 2015a; Qian et al., 2020; Talvitie & Singh, 2007; Serrano et al., 2021). From the perspective of what is being transferred, which means the known information, it can be generally divided into demonstrations (Ammar et al., 2015b; Shankar et al., 2022; Watahiki et al., 2023), policy (Wang et al., 2022; Yang et al., 2023; Gui et al., 2023; Chen et al., 2024), parameters (Devin et al., 2017; Zhang et al., 2021b), and value function (Torrey et al., 2008; Taylor et al., 2008).

Common practices to solve CDRL under different state and action representations include leveraging cycle consistency and transition between states and actions across two domains to discover mapping functions (Zhang et al., 2021a; You et al., 2022; Li et al., 2022; Wu et al., 2022; Raychaudhuri et al., 2021; Gui et al., 2023), or employing adversarial training techniques to identify mapping relationships between states and actions in the source and target domains (Gui et al., 2023; Li et al., 2022; Wulfmeier et al., 2017; Mounsif et al., 2020; Raychaudhuri et al., 2021; Watahiki et al., 2022).

## 3 PRELIMINARIES

In this section, we describe the standard problem formulation of preference-based RL. Throughout this paper, for any set $\mathcal{X}$, we use $\Delta(\mathcal{X})$ to denote the set of all probability distributions over $\mathcal{X}$.

**Markov Decision Processes.** As in typical RL, we model each domain as a Markov decision process (MDP) denoted by $\mathcal{M} = (\mathcal{S}, \mathcal{A}, \mathcal{T}, R, \mu, \gamma)$, where $\mathcal{S}$ and $\mathcal{A}$ denote the state space and action space, $\mathcal{T} : \mathcal{S} \times \mathcal{A} \to \Delta(S)$ is the transition kernel that maps each state-action pair to a probability distribution over the next state, $R : \mathcal{S} \times \mathcal{A} \to \mathbb{R}$ denotes the reward function, $\mu \in \Delta \mathcal{S}$ is the initial state distribution, and $\gamma \in (0, 1]$ is the discount factor. Let $\pi : \mathcal{S} \to \Delta(\mathcal{A})$ denote the policy of the RL agent and let $\tau = (s_0, a_0, r_1, \cdots)$ denote a trajectory generated under $\pi$ in the domain $\mathcal{M}$. Given a trajectory $\tau$, we slightly abuse the notation and use $R(\tau)$ to denote the total expected reward accrued along $\tau$, i.e., $R(\tau) := \sum_{t=0}^{\infty} R(s_t, a_t)$. Let $\Pi$ denote the set of all stationary Markov policies. We define the expected total discounted reward under $\pi$ as $V_{\mathcal{M}}^{\pi}(\mu) := \mathbb{E}[\sum_{t=0}^{\infty} \gamma^t R(s_t, a_t)|s_0 \sim \mu, \pi]$. Let $\pi_{\mathcal{M}}^* := \arg\max_{\pi \in \Pi} V_{\mathcal{M}}^{\pi}(\mu)$ be an optimal policy for $\mathcal{M}$ in that it maximizes the expected total discounted reward.

**Preference-based RL.** In the standard PbRL, the environment is modeled as an MDP $\mathcal{M} = (\mathcal{S}, \mathcal{A}, \mathcal{T}, R, \mu, \gamma)$ as usual. Moreover, the goal of PbRL remains the same as the standard reward-based RL, i.e., finding an optimal policy $\pi_{\mathcal{M}}^*$ that maximizes $V_{\mathcal{M}}^{\pi}(\mu)$. Despite the existence of an underlying true reward function (so that the RL objective function is well-defined), in the PbRL setting, the reward function $R$ is hidden and not observable to the learner during training. Nevertheless, given two trajectories $\tau$ and $\tau'$, the learner can receive the (possibly randomized) preference over $\tau$ and $\tau'$, which is determined by the total expected reward $R(\tau)$ and $R(\tau')$ along the trajectories. For notional convenience, we use $\tau \succ \tau'$ (or an equivalent expression $\tau' \prec \tau$) to denote the event that $\tau$ is preferred over $\tau'$. Note that a probability preference model $\mathcal{P}(\tau, \tau'; R)$ is typically needed to specify the likelihood of the event $\tau \succ \tau'$. For example, under the celebrated Bradley-Terry model (Bradley & Terry, 1952), we have $\mathcal{P}(\tau, \tau'; R) := 1/(1 + \exp(R(\tau') - R(\tau)))$. We assume that under the preference model, for any pair of trajectories $\tau, \tau'$, either the event $\tau \succ \tau'$ or $\tau' \succ \tau$ would happen at each time.

To solve PbRL, one popular way is to adopt a two-stage approach, where we first learn the underlying true reward function from the preference feedback and then apply an off-the-shelf RL algorithm

for policy learning. Under a preference model $\mathcal{P}(\tau, \tau'; R)$, a reward model $\hat{R}$ can be learned by maximizing the log-likelihood, i.e., given a dataset of trajectories $\mathcal{D}$, as Equation (1).

$$\hat{R} = \arg\max_{R':\mathcal{S} \times \mathcal{A} \to \mathbb{R}} \mathbb{E}_{\tau,\tau' \in \mathcal{D}, \tau \succ \tau'} \left[ \log \mathcal{P}(\tau, \tau'; R') \right]. \tag{1}$$

This approach has been widely used in the fine-tuning of large language models with RLHF (Ouyang et al., 2022). Additional related work on PbRL can be found in Appendix D.

## 4 PROBLEM FORMULATION

The proposed CD-PbRL problem extends the standard (unsupervised) CDRL problem, which aims to achieve knowledge transfer from a source domain to another target domain, to the scenario where the preferences over trajectories are available as weak supervision in the target domain. The source and target domains are modeled as follows:

**Source domain:** The source domain is modeled as an MDP denoted by $\mathcal{M}_{\text{src}}$ := $(\mathcal{S}_{\text{src}}, \mathcal{A}_{\text{src}}, \mathcal{T}_{\text{src}}, R_{\text{src}}, \mu_{\text{src}}, \gamma)$[1]. For efficient knowledge transfer, the source domain is typically an environment that is cheap and easy to access, *e.g.*, a simulator. Accordingly, we presume that the learner has full access to the source-domain environment and hence can collect data samples and obtain a pre-trained source-domain policy $\pi_{\text{src}}$. This setting has been adopted by most of the existing CDRL literature (Zhang et al., 2021a; Xu et al., 2023; Gui et al., 2023).

**Target domain:** Similarly, the target domain is modeled as an MDP denoted by $\mathcal{M}_{\text{tar}}$ := $(\mathcal{S}_{\text{tar}}, \mathcal{A}_{\text{tar}}, \mathcal{T}_{\text{tar}}, R_{\text{tar}}, \mu_{\text{tar}}, \gamma)$. Notably, the target-domain MDP can differ from source-domain MDP in *transition dynamics*, *state-action spaces*, etc., and we only assume that the two domains share the same discount factor, which is a fairly mild condition. In the standard unsupervised CDRL setting (Zhang et al., 2021a; Gui et al., 2023), the learner is given a set of target-domain trajectories $\mathcal{D}_{\text{tar}} = \{\tau_i\}_{i=1}^{D}$ collected under some behavior policy. Due to the unsupervised setting, the reward function $R_{\text{tar}}$ is assumed to be unobservable to the learner, and hence $\mathcal{D}_{\text{tar}}$ only contains information about the visited state-action pairs. Notably, this formulation can suffer from the identifiability issue by nature as described in Section 1. By contrast, built on the CDRL, our proposed CD-PbRL formulation additionally includes that the learner can further receive *preference information about pairs of trajectories* in the target domain, despite the unknown true rewards. The goal of CD-PbRL is to find an optimal policy $\pi^*_{\mathcal{M}_{\text{tar}}} := \arg\max_{\pi \in \Pi_{\text{tar}}} V^{\pi}_{\mathcal{M}_{\text{tar}}}(\mu_{\text{tar}})$ for the target domain.

## 5 METHODOLOGY

In this section, we formally present the proposed algorithm for the CD-PbRL problem. We start by describing the proposed CDPC principle and thereafter provide the implementation of the training and inference procedure of the resulting CDPC algorithm.

### 5.1 CROSS-DOMAIN PREFERENCE CONSISTENCY

To mitigate the correspondence identifiability issue, we propose to constrain the learning of state correspondence by *preference consistency*, which is meant to ensure that the preference ordering of the corresponding trajectories in the two domains remains consistent. An illustration of the CDPC principle is provided in Figure 2. To better motivate this, we can think of an analogy in language modeling: We

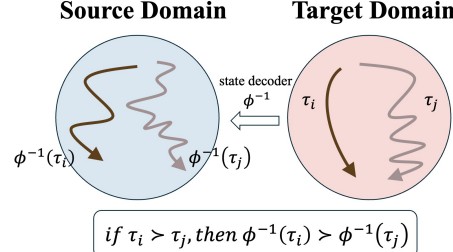

Figure 2: **The principle of cross-domain preference consistency**: Let $\tau_i$ and $\tau_j$ be two target-domain trajectories. If $\tau_i$ is preferred over $\tau_j$, which means it has a higher total return, then the trajectories transformed through a state decoder $\phi^{-1}$ shall maintain the same preference, i.e., $\phi^{-1}(\tau_i)$ shall be preferred over $\phi^{-1}(\tau_j)$.

can interpret $\tau_i$ and $\tau_j$ as two sentences written in German. The state decoder acts like a translator, converting a German sentence into one in English. If $\tau_i$ is more aligned with natural human language

---

[1]Throughout this paper, we use the subscripts "src" and "tar" to denote the objects of the source and the target domain, respectively.

in German than $\tau_j$, then after translation by the decoder, $\tau_i'$ is expected to be also more natural and fluent than $\tau_j'$ in English expression. The above characteristic can be used as an additional requirement to identify the inter-domain state correspondence.

Based on the concept of CDPC, here we provide an overview of the proposed algorithm, which consists of the following two major building blocks:

**Training phase: Learning a target-to-source state decoder by preference consistency.** As in typical CDRL methods, our CDPC framework also learns a state decoder $\phi^{-1} : \mathcal{S}_{\mathrm{tar}} \to \mathcal{S}_{\mathrm{src}}$ such that actions taken in $\mathcal{M}_{\mathrm{tar}}$ can be determined through knowledge transfer from a source-domain policy. Recall from Section 1 that fully unsupervised CDRL methods, where the state decoder is learned solely based on dynamics alignment (Gui et al., 2023) or reconstruction (Zhang et al., 2021a), can suffer from the identifiability issue. As a result, we propose to learn the state decoder based on the CDPC principle, which serves as an additional criterion for learning the state correspondence across domains. Specifically, to learn the state decoder[2] $\phi_\theta^{-1} : \mathcal{S}_{\mathrm{tar}} \to \mathcal{S}_{\mathrm{src}}$ (parameterized by $\theta$), we construct a cross-domain loss function based on the pairwise ranking idea in PbRL as follows:

$$\mathcal{L}_{\mathrm{pref}}(\theta) := \mathbb{E}_{\tau_i, \tau_j \sim \mathcal{D}_{\mathrm{tar}}} \left[ \log \left( 1 + e^{R_{\mathrm{src}}\left(\phi_\theta^{-1}(\tau_j)\right) - R_{\mathrm{src}}\left(\phi_\theta^{-1}(\tau_i)\right)} \right) \right]. \tag{2}$$

The preference loss function in Equation (2) resembles Equation (1) of PbRL but with one major difference: the preference consistency is captured through the state decoder $\phi_\theta^{-1}$. This preference loss function can be used in conjunction with any other off-the-shelf loss function for unsupervised CDRL, such as dynamics cycle consistency or reconstruction loss (Zhang et al., 2021a). More implementation details of the state decoder are described in Section 5.2.

**Inference phase: Selecting target-domain actions by MPC in target domain with cross-domain trajectory optimization.** With a properly learned state decoder, the next step is to transfer the pre-trained source-domain policy $\pi_{\mathrm{src}}$ to the target domain. Notably, one naive approach is to simply learn an additional action encoder $\psi : \mathcal{A}_{\mathrm{src}} \to \mathcal{A}_{\mathrm{tar}}$ (e.g., similarly by preference consistency) such that given any state $s \in \mathcal{S}_{\mathrm{tar}}$, a target-domain action can be induced by $\psi(a_{\mathrm{src}})$ with $a_{\mathrm{src}} \sim \pi_{\mathrm{src}}(\phi^{-1}(s))$, as also adopted by Gui et al. (2023). However, this approach can suffer from inaccurate preference correspondence. The details about this naive approach are provided in Appendix B.

To better leverage the CDPC principle in selecting actions in the target domain, we propose to enforce knowledge transfer from the perspective of *planning*. Specifically, we use MPC in the target domain with the help of *cross-domain trajectory optimization* (CDTO). The detailed implementation is provided in Section 5.3.

## 5.2 TRAINING PHASE OF CDPC: LEARNING A STATE DECODER

In the CD-PbRL setting, a well-trained state decoder $\phi_\theta^{-1}$ should satisfy the following characteristics: (i) $\phi_\theta^{-1}$ shall be able to ensure preference consistency between trajectories and (ii) meet the original cycle consistency conditions in both state construction and dynamics alignment. To learn the state decoder, we use the preference consistency loss as described in Section 5.1 as well as the dynamics cycle consistency loss and reconstruction loss.

**Dynamics Cycle Consistency Loss:** One common principle of learning state-action correspondence is through dynamics alignment, i.e., the next state obtained by the state decoder shall be consistent with that generated under the source-domain transition dynamics. Specifically, in this work, we use the following loss function to capture dynamics cycle consistency:

$$\mathcal{L}_{\mathrm{dcc}}(\theta) := \mathbb{E} \left[ \left\| \mathcal{T}_{\mathrm{src}} \left( \phi_\theta^{-1}(s), a \right) - \phi_\theta^{-1}(s') \right\|^2 \right], \tag{3}$$

where the expectation is over the randomness of $s, s' \sim \mathcal{D}_{\mathrm{tar}}$ and $a \sim \pi_{\mathrm{src}}(\cdot | \phi^{-1}(s))$, and $\mathcal{T}_{\mathrm{src}}$ is directly accessible.

**Reconstruction Loss:** Additionally, the reconstruction loss (Zhang et al., 2021a; Gui et al., 2023; Zhu et al., 2017) is widely used in cross-domain tasks for its several advantages: (i) It acts as a regularization term, encouraging the decoder to produce outputs closely resembling the input data.

---

[2]Here we use the term "decoder" as this mapping function is typically learned based on an autoencoder network architecture.

**Algorithm 1** Cross-Domain Preference Consistency (CDPC)

**Require:**
 A dataset of target-domain trajectories $\mathcal{D}_{\text{tar}}$
1: **for** each episode $k$ **do**
2:   // Training
3:   Sample $\tau_i, \tau_j \sim \mathcal{D}_{\text{tar}}$
4:   Obtain the preference label for $\tau_i, \tau_j$
5:   Update state decoder $\phi_\theta^{-1}$ by taking a gradient step based on $\mathcal{L}_{\text{total}}(\theta)$ (Equation (5))
6:   // Validation
7:   **for** each timestep $t$ **do**
8:     $s_t \leftarrow$ current state in the target domain environment
9:     Select optimal action $a_t$ using Algorithm 2
10:     Apply $a_t$ to the target-domain environment
11:   **end for**
12: **end for**

**Algorithm 2** Cross-Domain Trajectory Optimization (CDTO)

**Require:**
 state $s_t$, state decoder $\phi^{-1}$
**Ensure:**
 action $a_t$
1: Initialize $\mathcal{D}^{(t)} \leftarrow \emptyset$
2: Generate synthetic trajectories $\tau_{1:m}$ using policy network $\pi_\iota(s)$ and dynamics model $F_\gamma(s, a)$
3: $\mathcal{D}^{(t)} \leftarrow \mathcal{D}^{(t)} \cup \{\tau_1, \tau_2, ..., \tau_m\}$
4: Decode $\tau_{1:m}$ using state decoder $\phi_\theta^{-1}$
5: Compute $R_s^{1:m}$ using source-domain reward function $R_{src}$
6: Sort $\tau_{1:m}$ by $R_s^{1:m}$ in descending order
7: $\tau^* \leftarrow \mathcal{D}^{(t)}[0]$
8: $a^* \leftarrow$ first action of $\tau^*$
9: return $a^*$

This enhances reconstruction quality and generalization across domains. (ii) The loss fosters model stability by promoting consistency between input and reconstructed outputs, even in the presence of noise or domain variations. Minimizing the reconstruction loss leads to a more compact and meaningful data representation, facilitating better transfer learning and generalization capabilities. The reconstruction loss is defined as

$$\mathcal{L}_{\text{rec}}(\theta) := \mathbb{E}\left[\left\|\phi_\omega\left(\phi_\theta^{-1}(s)\right) - s\right\|^2\right], \tag{4}$$

where the expectation is over the randomness of the state $s$ drawn from the target-domain dataset $\mathcal{D}_{\text{tar}}$. Note that we presume the use of an autoencoder, where $\phi$ and $\omega$ represent the parameters of the state decoder and encoder, respectively. As we only need the decoder for inference, we ignore the dependency of $\mathcal{L}_{\text{rec}}(\theta)$ on $\omega$ in Equation (4) for brevity.

In summary, the total loss of the state decoder can be expressed as follows:

$$\mathcal{L}_{\text{total}}(\theta) := \mathcal{L}_{\text{pref}}(\theta) + \beta_1 \mathcal{L}_{\text{dcc}}(\theta) + \beta_2 \mathcal{L}_{\text{rec}}(\theta), \tag{5}$$

where $\beta_1 > 0$ and $\beta_2 > 0$ are the weights for balancing the three loss terms. The overall pseudocode is provided in Algorithm 1.

### 5.3 INFERENCE PHASE OF CDPC: CROSS-DOMAIN MPC

During the inference phase, given a well-trained state decoder, we propose to determine target-domain actions through planning via cross-domain MPC, which consists of two major components:

**Cross-domain trajectory optimization (CDTO)**: As in typical MPC, at each time step $t$, based on the current observation $s_t$, we determine the action $a_t$ by (i) generating multiple synthetic trajectories of length $h$ with $s_t$ as the starting state (denoted by $\mathcal{D}^{(t)}$) in the target domain, and then (ii) selecting one trajectory $\tau$ from $\mathcal{D}^{(t)}$ based on some performance metric, and (iii) choosing the first action of $\tau$ as the action $a_t$. Notably, to implement (ii), we propose to use the source-domain reward of the source-domain trajectory induced by the state decoder as the selection criterion for MPC.

**Generation of synthetic trajectories for cross-domain MPC**: To implement the subroutine (i) in CDTO, we also learn two helper models based on the target-domain dataset $\mathcal{D}_{\text{tar}}$, namely a target-domain dynamics model (learned in a standard way by minimizing squared errors of next-state prediction) and a target-domain policy by behavior cloning. This can be viewed as a variant of the random shooting technique in the model-based RL literature (Nagabandi et al., 2018; 2020) but with a behavior-cloned policy.

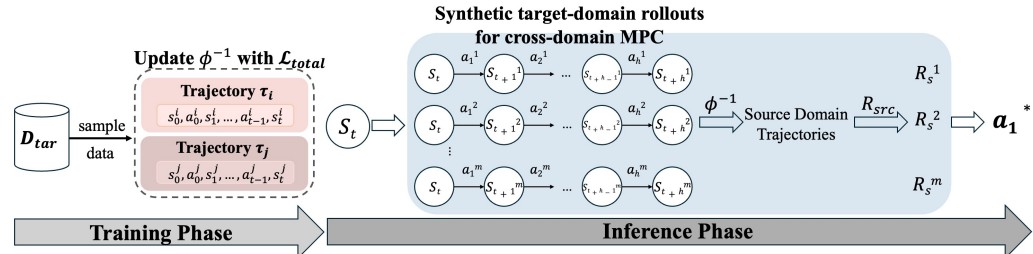

Figure 3: **An illustration of cross-domain MPC:** During inference, based on the current state $s_t$, we generate $m$ synthetic trajectories of length $h$ by using a learned target-domain dynamics model and utilizing a behavior-cloned policy $\pi_\iota$ from $\mathcal{D}_{\text{tar}}$. These $m$ trajectories are then mapped into the corresponding source trajectories using the trained state decoder $\phi_\theta^{-1}$. We compute the total return for each trajectory separately using the source-domain reward function (available in the cross-domain RL setting). Finally, the first action $a_1^*$ from the sequence with the highest total return is adopted.

The cross-domain MPC approach is illustrated in Figure 3. Note that here we choose the most basic variant of MPC during inference mainly to show the effectiveness of CDPC framework. The proposed framework can be readily enhanced and integrated with more sophisticated MPC methods, such as the popular cross-entropy method (Botev et al., 2013) and the filtering and reward-weighted refinement (Nagabandi et al., 2020). The overall pseudocode is provided in Algorithm 2.

# 6 EXPERIMENTAL RESULTS

## 6.1 EXPERIMENTAL CONFIGURATION

**Environment domains.** We utilize MuJoCo and Robosuite to simulate robot locomotion and manipulation, respectively. While MuJoCo and Robosuite already have pre-configured reward functions, given the CD-PbRL problem setting, we will not utilize them during training; they will only serve as performance metrics for evaluation.

- **MuJoCo.** We consider three MuJoCo tasks, namely Reacher, HalfCheetah, and Walker. Regarding the cross-domain setting, we use the original MuJoCo environments as the source domains and consider robots of more complex morphologies (and hence with higher state and action dimensionalities) as the target domains, The detailed description about the source domain and target domain can be found in Table 1 and Figure 4.

- **Robosuite.** We set the source domain and target domain as two structurally different robot arms, namely Panda and IIWA, which have distinct state-action representations. We let the two types of robot arms perform the same set of tasks, including Lift, Door, and Assembly. The detailed description of the source domain and target domain can be found in Table 2 and Figure 4. All of the detailed information about the environments is provided in Appendix C.

**Benchmark methods.** We compare CDPC with multiple benchmark algorithms, including:

- **CAT-TR:** CAT is a CDRL method proposed by You et al. (2022) that learns state-action correspondence incorporating PPO using the true target-domain environmental reward. This robust use of information is expected to lead to better performance compared to CDPC.

- **Dynamics Cycle-Consistency (DCC):** DCC is an unsupervised CDRL method (Zhang et al., 2021a) that learns state-action correspondence by cycle consistency in dynamics and reconstruction. We use DCC as a baseline since both DCC and CDPC learn without knowing the true target-domain environmental rewards.

- **Cross-Morphology Domain Policy Adaptation (CMD):** CMD is a more recent unsupervised CDRL method (Gui et al., 2023) specifically for transfer in cross-morphology problems. CMD also serves as a suitable baseline since both CMD and CDPC are designed to learn without knowing the true target-domain environmental rewards.

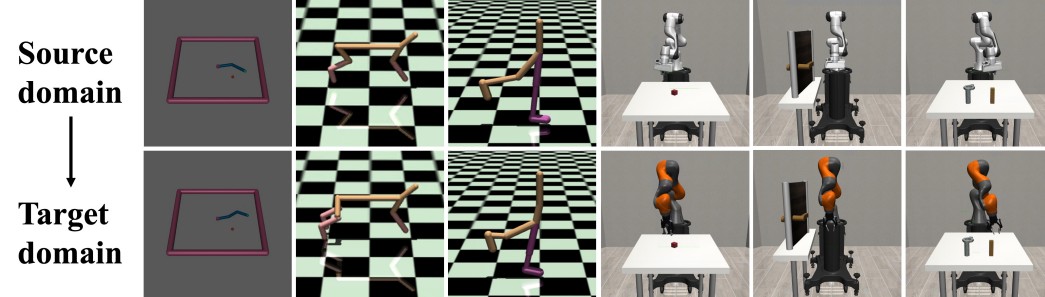

Figure 4: **Agent morphologies of the source domain and the target domain in MuJoCo and Robosuite:** The top row represents the source domain, which includes: Reacher, Halfcheetah, Walker, Panda-Lift, Panda-Door, Panda-NutAssembleRound. The bottom row represents the target domain, which includes: Reacher-3joints, Halfcheetah-3legs, Walker-head, IIWA-Lift, IIWA-Door, and IIWA-NutAssembleRound.

- **SAC-Off-TR:** This method employs offline SAC directly with target-domain data, without using transfer learning. By leveraging true target-domain environmental rewards, it serves as a natural and expectedly strong benchmark method, even without transfer learning.

- **SAC-Off-RM:** Compared to SAC-Off-TR, this method uses a reward model trained with RLHF loss (Memarian et al., 2021) instead of the true target-domain environmental reward. This approach allows us to directly compare the effectiveness of using preferences, as in CDPC, with the alternative of learning a reward model from preferences first.

- **% BC:** Behavior cloning using the top $X\%$ of the trajectories in the dataset $\mathcal{D}_{\text{tar}}$, where $X \in \{10\%, 20\%, 50\%\}$. We will use this as a baseline because we can convert the concept of pairwise preference into ranking within $\mathcal{D}_{\text{tar}}$.

**Dataset.** As described in the problem formulation of CD-PbRL, a target-domain dataset $\mathcal{D}_{\text{tar}}$ is provided to the learner. To implement this, we follow the data collection method of D4RL (Fu et al., 2020). Specifically, we mix the expert demonstrations (by an expert policy learned under SAC (Haarnoja et al., 2018)) and sub-optimal data generated by unrolling a uniform-at-random policy. The size of $\mathcal{D}_{\text{tar}}$ for each task is provided in Appendix E. For the main experiments, the proportion of expert trajectories in the dataset is set to be 20%. For a fair comparison, this dataset is shared by all algorithms in the experiments. An empirical study on the mixing proportion is provided in the sequel.

More details about the experimental configuration can be found in Appendix C.2.

## 6.2 RESULTS AND DISCUSSIONS

**Does CDPC achieve data-efficient cross-domain transfer in RL?** The results of final total rewards are shown in Figure 5, indicating that CDPC converges faster and performs better than the baselines. The reason why DCC and CMD perform relatively poorly is that they suffer from the identifiability issue as they only focus on learning the state-action correspondence between two domains. SAC-Off-RM, on the other hand, needs to first learn a reward model, and if the reward model is inaccurate, it greatly impacts the results. SAC-Off-TR converges more slowly as it does not involve any knowledge transfer from the source domain.

**Does CDPC learn an effective state decoder $\phi^{-1}$?** We compare CDPC with other CDRL benchmark methods in the effectiveness of the learned state decoders. The results of final total rewards are shown in Figure 6, indicating that CDPC converges faster and achieve higher total rewards than all the other CDRL methods, even than CAT-TR with true reward signals.

**Does CDPC learn a state decoder that can effectively achieve cross-domain preference consistency?** We provide an ablation study and investigate the significance of the preference consistency loss. The results showed that the preference consistency loss has a highly significant effect. Without using $\mathcal{L}_{\text{pref}}(\theta)$, the decoder encounters identifiability issues, making it unable to decode good trajectories into corresponding source trajectories. Consequently, it also becomes unable to utilize the MPC module to select suitable actions. The results are shown in Figure 7. We also provide a Reacher example for visualization (with the link provided in Appendix E).

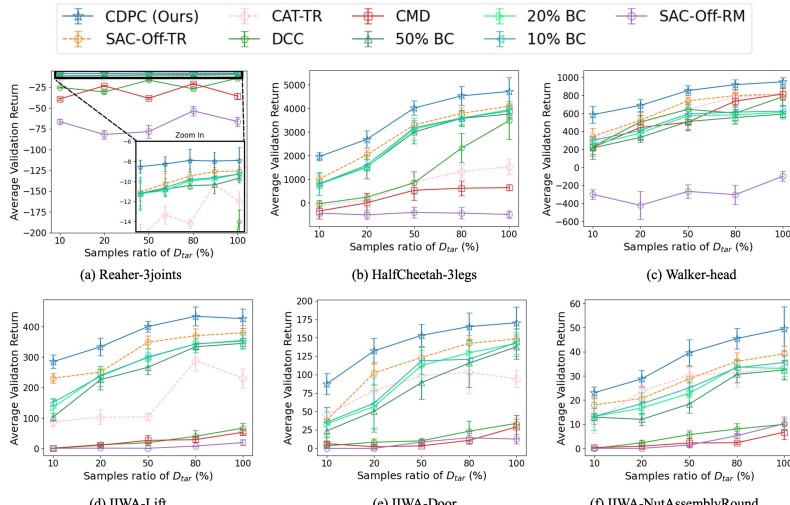

Figure 5: **Sample efficiency of CDPC and the benchmark methods:** CDPC demonstrates greater efficiency compared to the baseline methods across various dataset sizes, maintaining strong performance even as the dataset scale increases.

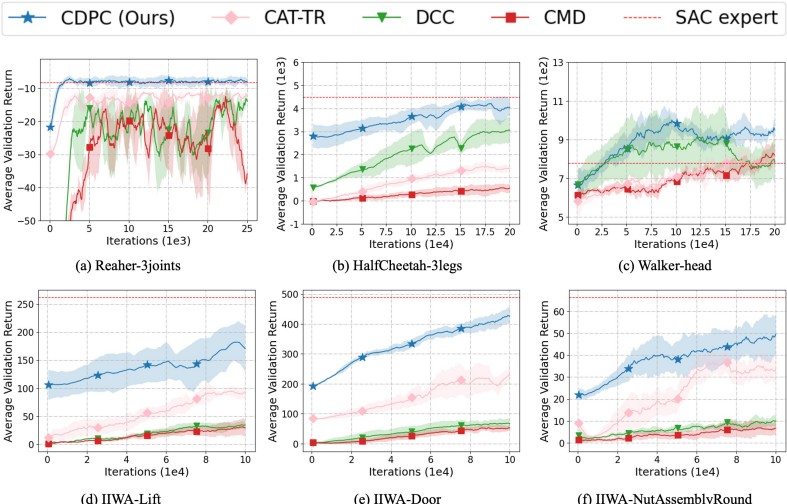

Figure 6: **Decoder performance of CDPC and the benchmark methods:** The learning curve of the CDPC decoder demonstrates a consistent improvement over the baseline methods, particularly in terms of convergence speed and final performance.

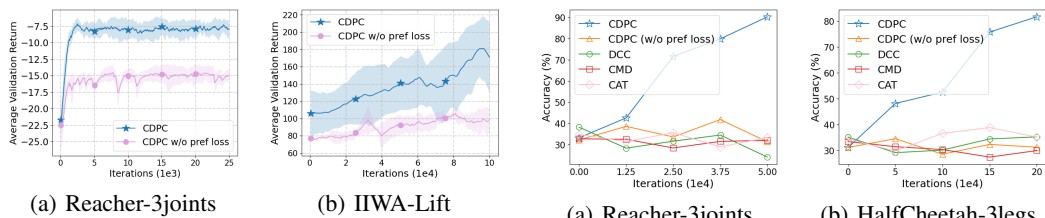

Figure 7: **Learning curves of CDPC with and without the preference consistency loss**: The decoder trained with preference consistency loss yields noticeably improved results compared to the one trained without this loss.

Figure 8: **Preference accuracy of the state decoders learned by CDPC, DCC, CMD, and CAT:** The integration of preference consistency loss enables CDPC to attain higher preference accuracy than the baseline methods.

Moreover, we also compare the state decoders learned by CDPC, DCC, and CMD in terms of their capabilities to maintain preference consistency across domains. The results, as shown in Figure 8, indicate that the CDPC decoder is significantly better in achieving preference consistency.

**Does the quality of the target-domain data have a significant impact on CDPC?** Recall that CDPC learns from a target-domain $\mathcal{D}_{tar}$ with mixed samples collected by an expert policy and a uniform-at-random policy. Let $\alpha \in [0, 1]$ denote the mixture proportion of expert data. We evaluate CDPC under four choices of mixture proportions and observe that CDPC is not very sensitive to the data quality. The results are shown in Figure 9. Even without any expert data, the performance of CDPC remains competitive compared to the baselines.

**Does the quality of preference labels have a significant impact on CDPC?** We experimented with flipping 10%, 20%, and 50% of the preference labels and found that CDPC still can learn successfully when only a certain proportion of the preference labels are scrambled, as shown in Figure 10.

**How is the cross-domain transferability of CDPC under different domain similarities?** To answer this, we constructed variants of Reacher environments with 4 joints, 5 joints, and 6 joints as the target domains and take the vanilla Reacher with 2 joints as the source domain. From Figure 11, we observe that CDPC can still reliably achieve cross-domain transfer despite the slight decrease in the transfer performance with the number of joints. Notably, without the true target-domain reward signal, CDPC can still achieve comparable or better cross-domain performance than CAT-TR, which has access to the target-domain true reward.

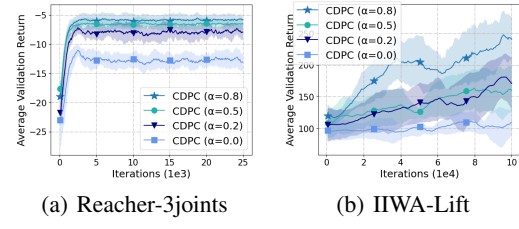

(a) Reacher-3joints     (b) IIWA-Lift

Figure 9: **Learning curves of CDPC under different mixing rates of expert data** $\alpha$**:** CDPC can benefit from a higher proportion of expert data and perform reliably with limited or no expert data.

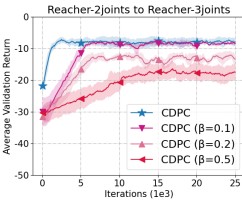

Figure 10: **Learning curves of CDPC under different flipping ratios of preference label** $\beta$**:** Even with flipping applied to some preference labels, CDPC can still achieve successful transfer.

## 7 CONCLUSION

We study CD-PbRL, a new cross-domain RL problem with preference feedback, and propose a generic CDPC framework that enforces preference alignment between the source and target domains. Based on this concept, we propose the CDPC algorithm that combines a state decoder learned by preference consistency loss for training and a cross-domain MPC method for inference. Through extensive experiments on various robotic tasks, we confirm that CDPC indeed serves as a promising solution to achieving effective and data-efficient cross-domain transfer across domains.

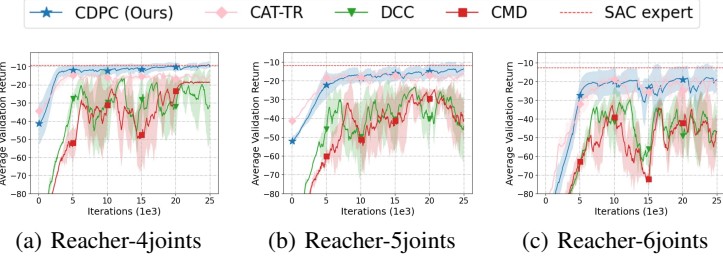

(a) Reacher-4joints     (b) Reacher-5joints     (c) Reacher-6joints

Figure 11: **Learning curves of CDPC under different domain similarities:** As the domain dissimilarity between the source and target domains increases, successful transfer becomes more difficult. Nevertheless, CDPC maintains a performance advantage over the baseline methods.

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

APPENDICES

# Table of Contents

## A  A DETAILED DESCRIPTION OF THE MOTIVATING EXAMPLE IN FIGURE 1

Here, we explain the detailed steps of the gridworld example in Figure 1.

**Problem setup:** Consider one target domain trajectory $\tau$, two state decoder $\phi_\alpha^{-1}$ and $\phi_\beta^{-1}$, one well-trained source domain policy $\pi_{src}$, source domain reward function $R_{src}$, which is defined as follows: the top-left corner is the starting point, the bottom-left corner contains the treasure, which provides a reward of +0.5 upon reaching it, and the bottom-right corner is the goal, which provides a reward of +1 and terminates the episode. For simplicity, let us assume discount factor $\gamma$ equals to 1.

For $\phi_\alpha^{-1}$, the process of decoding can be described as follows:

1. $\phi_\alpha^{-1}(00, 00) \Rightarrow (0, 0)$, $\pi_{src}(0, 0) = \rightarrow$, go to $(0, 1)$, reward = +0
2. $\phi_\alpha^{-1}(00, 01) \Rightarrow (0, 1)$, $\pi_{src}(0, 1) = \rightarrow$, go to $(0, 2)$, reward = +0
3. $\phi_\alpha^{-1}(00, 10) \Rightarrow (0, 2)$, $\pi_{src}(0, 2) = \downarrow$, go to $(1, 2)$, reward = +0
4. $\phi_\alpha^{-1}(01, 10) \Rightarrow (1, 2)$, $\pi_{src}(1, 2) = \downarrow$, go to $(2, 2)$, reward = +1
5. $\phi_\alpha^{-1}(10, 10) \Rightarrow (2, 2)$, total return = 1

For $\phi_\beta^{-1}$, the process of decoding can be described as follows:

1. $\phi_\beta^{-1}(00, 00) \Rightarrow (0, 0)$, $\pi_{src}(0, 0) = \downarrow$, go to $(1, 0)$, reward = +0
2. $\phi_\beta^{-1}(00, 01) \Rightarrow (1, 0)$, $\pi_{src}(1, 0) = \downarrow$, go to $(2, 0)$, reward = +0.5
3. $\phi_\beta^{-1}(00, 10) \Rightarrow (2, 0)$, $\pi_{src}(2, 0) = \Rightarrow$, go to $(2, 1)$, reward = +0
4. $\phi_\beta^{-1}(01, 10) \Rightarrow (2, 1)$, $\pi_{src}(2, 1) = \rightarrow$, go to $(2, 2)$, reward = +1
5. $\phi_\beta^{-1}(10, 10) \Rightarrow (2, 2)$, total return = 1.5

However, we cannot determine whether $\tau'_\alpha$ or $\tau'_\beta$ is better, without considering total return. As a result, it remains infeasible to distinguish between them if we only use dynamic cycle consistency loss. Without a suitable mechanism for choosing between $\phi_\alpha^{-1}$ or $\phi_\beta^{-1}$, the correspondence identifiability issue could easily arise.

## B  DISCUSSION: A NAIVE CD-PBRL APPROACH WITH AN ACTION ENCODER

The most naive approach to addressing inter-task mapping problems is to train mapping functions for both state and action. A simple illustration and explanation are provided in Figure 12. Initially, we employed the concept of preference consistency to train an autoencoder for both state and action. However, the results were highly unstable, and since there was no information available regarding the target domain's reward, we needed to additionally train a reward model in the target domain to ensure both domains had preference information to maintain bidirectional mapping. A particularly tricky aspect is that if the reward model is not well-trained easily, the preference labels provided by the reward model will be incorrect, which will lead to poor performance of the action encoder. We also included the training results of this naive method in Figure 12.

Finally, we cleverly combined the preference consistency state decoder with MPC, which only required finding a decoder that could ensure consistent preferences, guaranteeing the effectiveness of the MPC approach.

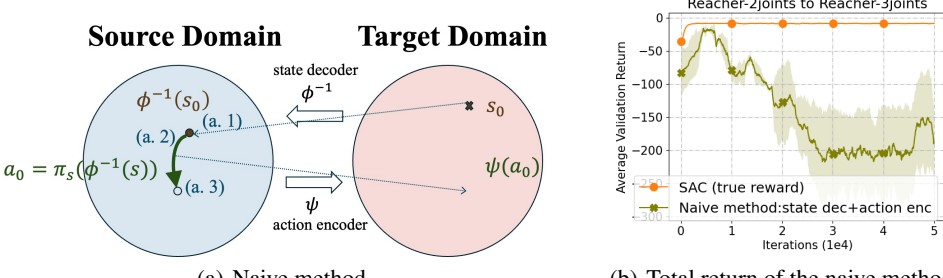

(a) Naive method                                  (b) Total return of the naive method

Figure 12: **Naive method:** (a) (a.1)First, the target state is transformed into the corresponding source state through the decoder. (a.2)Second, Using the known source domain policy, an action is selected in the source domain. (a.3)Finally, the action encoder transforms this action into the corresponding target action to complete one step. This process is repeated until termination. (b) Performance of naive method is poor and unstable.

## C  DETAILED EXPERIMENTAL CONFIGURATIONS

### C.1  DETAILED CONFIGURATIONS OF ENVIRONMENTS

Table 1: **Differences between source and target domain in MuJoCo**

|        |            | Reacher | HalfCheetah | Walker |
|--------|------------|---------|-------------|--------|
| **Source** | state dim  | 11      | 17          | 17     |
| **Domain** | action dim | 2       | 6           | 6      |
| **Target** | state dim  | 12      | 23          | 19     |
| **Domain** | action dim | 3       | 9           | 7      |

Table 2: **Differences between source and target domain in Robosuite**

|        |            | Lift | Door | NutAssemblyRound |
|--------|------------|------|------|------------------|
| **Source** | state dim  | 42   | 46   | 46               |
| **Domain** | action dim | 7    | 7    | 7                |
| **Target** | state dim  | 50   | 54   | 54               |
| **Domain** | action dim | 7    | 7    | 7                |

The detailed descriptions of the environments of our experiments are as follows:

- **Reacher:** MuJoCo Reacher is an environment commonly used in reinforcement learning research. In this environment, an agent, typically a robotic arm, must learn to control its movements to reach a target location. The agent receives observations such as position and velocity of its joints, and its goal is to learn a policy that enables it to efficiently navigate its arm to the target.

- **HalfCheetah:** MuJoCo HalfCheetah is a simulated environment frequently utilized in reinforcement learning research. In this environment, an agent, typically a virtual half-cheetah, learns to navigate and control its movements in a physics-based simulation. The primary objective for the agent is to achieve efficient locomotion while adhering to physical constraints. The HalfCheetah environment offers a continuous control task, where the agent must learn to balance speed and stability to achieve optimal performance.

- **Walker:** MuJoCo Walker is a simulated environment frequently utilized in reinforcement learning research. In this environment, an agent, typically a virtual bipedal walker, learns to navigate and control its movements in a physics-based simulation. The primary objective for the agent is to achieve efficient and stable bipedal locomotion while adhering to physical constraints. The Walker environment offers a continuous control task, where the agent must learn to balance, walk, and sometimes recover from disturbances to achieve optimal performance.

- **Panda:** RoboSuite Panda is a versatile robotic platform featuring a highly dexterous Panda robot arm. It's designed for research and development in robotics, offering flexibility for various tasks like manipulation and assembly. With its user-friendly interface and comprehensive software framework, it fosters innovation and collaboration in both academic and industrial settings. Our experimental tasks include Block Lifting, Door Opening, and Nut Assembly Round.

- **IIWA:** RoboSuite IIWA presents an advanced robotic platform centered around the highly sensitive and versatile IIWA robotic arm. Tailored for research and development, it excels in precision tasks like assembly and pick-and-place operations. Its intuitive interface and robust software framework support experimentation with cutting-edge robotics algorithms. Whether in academia or industry, RoboSuite IIWA empowers users to explore the forefront of robotic technology.

## C.2 EXPERIMENTAL SETUP

**Device.** CPU AMD Ryzen 9 7950X 32 threads, GPU NVIDIA GeForce RTX 4080, RAM 64GB DDR5, Storage 2TB NVMe SSD.

**Codebase.** For the implementation of SAC, we follow the GitHub codebase: `https://github.com/quantumiracle/Popular-RL-Algorithms/tree/master`. For the implementation of Robosuite policy, we follow the GitHub codebase: `https://github.com/ARISE-Initiative/robosuite-benchmark/tree/master`. For the implementation of DCC and CMD, we follow the GitHub codebase: `https://github.com/sjtuzq/Cycle_Dynamics/tree/master`. For the implementation of CAT, we follow the GitHub codebase:`https://github.com/TJU-DRL-LAB/transfer-and-multi-task-reinforcement-learning/tree/main/Single-agent%20Transfer%20RL/Cross-domain%20Transfer/CAT`.

**Hyperparameters.** We train source domain policy using SAC for 1e6 episodes, 128 for batch size, 3e-4 for Q network, policy and alpha learning rate. Target domain expert policy using SAC for 500 episodes, 128 for batch size, 3e-4 for Q network, policy and alpha learning rate. Decoder using LSTM for batch size 32, 1e-3 for learning rate run for 5 random seeds. The size of $\mathcal{D}_{\text{tar}}$ is 5e5 transition pairs for all tasks.

## C.3 EVALUATION OF PREFERENCE ACCURACY IN FIGURE 8

We provide the detailed procedure of the evaluation of preference accuracy used by CDPC and other benchmark methods in Figure 8 as follows:

- Step 1: Collect a target-domain dataset $\mathcal{D}'_{\text{tar}}$ of trajectories with preference labels.

- Step 2: Randomly sample a batch of $k$ trajectory pairs $\{(\tau_1^{(i)}, \tau_2^{(i)})\}_{i=1}^k$ and the corresponding preference label $y^{(i)}$ from $\mathcal{D}'_{\text{tar}}$. Feed each pair $(\tau_1^{(i)}, \tau_2^{(i)})$ into the learned state decoder

$\phi^{-1}$ and get the corresponding source-domain trajectories $(\tau_1^{(i)\prime}, \tau_2^{(i)\prime})$. Accordingly, let $z^{(i)}$ denote the source-domain preference label of $(\tau_1^{(i)\prime}, \tau_2^{(i)\prime})$.

- Step 3: Compute Accuracy = $\frac{\sum_{i=1}^{k} \mathbb{I}\{y^{(i)} = z^{(i)}\}}{k} \times 100\%$.

## D EXTENDED RELATED WORK

**Preference-based RL (PbRL).** PbRL (Wirth et al., 2017; Busa-Fekete & Hüllermeier, 2014; Kamishima et al., 2010; Wirth & Fürnkranz, 2013; Choi et al., 2024; Singh et al., 2024; Cheng et al., 2024) is a popular RL setting that focuses on learning policies or value functions from preferences rather than explicit reward signals. One common approach is to model the preference feedback as a binary classification problem (Lee et al., 2021a;b; Akrour et al., 2011; Pilarski et al., 2011; Akrour et al., 2012; Wilson et al., 2012; Ibarz et al., 2018). PbRL has been applied to various real-world domains, including personalized recommendation systems (Li et al., 2010), interactive learning from human feedback (Knox & Stone, 2009), and robot learning from human preferences (Warnell et al., 2018). Besides, PBRL can also be employed for automatic summarization of articles (Stiennon et al., 2020). This approach enables the model to acquire sophisticated summarization techniques through preference-based learning (Stiennon et al., 2020; Ouyang et al., 2022; Achiam et al., 2023; Lee et al., 2023; Kirk et al., 2023; Sun et al., 2023a). Beyond its application in large language models, preference-based techniques are also commonly utilized in training RL agents (Memarian et al., 2021; Liu et al., 2023; Chakraborty et al., 2023; Sun et al., 2023b). By leveraging human feedback to train reward functions, these techniques enable RL agents to approximate real-world rewards more accurately, guiding the agents towards convergence to an optimal policy.

## E VIDEOS

The link to the video is `https://imgur.com/a/cdpc-decoder-visualization-KvzLOqA`. A clarification is warranted regarding the observation that the target point in the decoded trajectory continues to shift, while the robotic arm exhibits minimal movement. This is because our decoder takes the entire state as input, and the target point position is included in the state. Practically, it's challenging to ensure that the decoded target point position remains the same each time. However, in the Reacher environment, a trajectory can be considered good if the total distance between the fingertip position and the target point position is minimized throughout the episode. The decoder ensures that the decoded trajectory maintains preference consistency, and we can leverage this characteristic with MPC to select the optimal actions.

## F ADDITIONAL EXPERIMENTS

### F.1 AN EMPIRICAL STUDY ON THE EFFECT OF DYNAMICS MODEL QUALITY ON CDPC PERFORMANCE

In this section, we conduct an additional empirical study to evaluate the robustness of CDPC to the quality of the learned dynamics model. To showcase this, we add additional perturbation noise to the predicted states output by the dynamics model. Intuitively, one shall expect that the decision made by the MPC procedure can be affected by the perturbation noise. Specifically, we first generate Gaussian random variables with zero mean and a standard deviation of $\alpha$. Based on the state representations provided by the official MuJoCo and Robosuite documentation, the noise terms are further rescaled according to the range of each dimension of the state. The experimental results are provided in the table below. We can observe that despite the lowered quality of the dynamics model, the performance of CDPC is only slightly affected and still remains fairly robust and superior to the strong benchmark SAC-Off-TR, which learns directly from the true target-domain reward function.

Table 3: **Performance comparison of CDPC under a noisy dynamics model under different perturbation magnitudes** $\alpha$**:** We can observe that despite the noisy dynamics model, the performance of CDPC is only slightly affected and still remains fairly robust and superior to the strong benchmark SAC-Off-TR, which learns directly from the true target-domain reward function.

| $\alpha$ | Reacher | IIWA-Lift |
|---|---|---|
| **0.0** | -7.9 ± 1.29 | 170.45 ± 21.49 |
| **0.1** | -8.05 ± 1.32 | 166.23 ± 20.09 |
| **0.2** | -8.31 ± 1.21 | 162.01 ± 22.06 |
| **0.4** | -8.82 ± 1.57 | 158.67 ± 18.35 |
| **0.8** | -9.21 ± 1.43 | 152.66 ± 18.85 |
| **SAC-Off-TR** | -8.97 ± 0.43 | 148.44 ± 13.24 |

### F.2 COMPARISON OF CDPC AND MPC-BASED BASELINES

In this section, we further demonstrate that the empirical strength of the CDPC algorithm indeed mostly come from the design of cross-domain preference consistency. To address this, we further compare CDPC in two environments, namely Reacher and IIWA-Lift, with three additional baselines as follows:

- **MPC:** This method employs MPC directly in the target domain, without using transfer learning. Here, we use the same dynamics model for both the pure MPC method and CDPC. The purpose of including baseline is to verify whether CDPC performs well simply because MPC itself is inherently strong.

- **CAT-TR-MPC:** Regarding CAT (You et al., 2022) mentioned in Section 6, we remove CAT's original action encoder and instead use MPC to select actions, similar to CDPC. Here, the main purpose is to verify whether the integration of MPC and other cross-domain RL methods (like CAT) already achieves strong empirical performance.

- **DCC-MPC:** Similar to CAT-TR-MPC, DCC-MPC is another baseline that integrates (Zhang et al., 2021a) with the MPC subroutine for taregt-domain action selection. Again, the main purpose here is to check whether the integration of MPC and other cross-domain method like DCC already achieves good empirical performance.

We report the experimental results on the sample efficiency, decoder performance, preference accuracy in Figure 13, Figure 14, and Figure 15, respectively. We can make several observations from these results:

- **CDPC is indeed more sample-efficient that pure target-domain MPC**: CDPC still remains best after the three MPC-based baselines are included. Notably, using MPC directly in the target domain can produce decent actions, resulting in a moderately high total return. However, pure target-domain MPC still underperforms CDPC since CDPC, as a cross-domain transfer method, nicely leverages the learned model from the source domain.

- **CAT-TR-MPC and DCC-MPC suffer from low preference accuracy and hence do not perform well**: On the other hand, CAT-TR-MPC and DCC-MPC completely fail to learn. This is because the state decoders of these methods are still not able to produce correct trajectory rankings even under the integration with the MPC module. This issue is particularly evident from the accuracy charts provided in Figure 15.

Based on the above, we conclude that the empirical strength of CDPC does not rely solely on MPC; rather, the key is the seamless integration of the preference-based state decoder with the cross-domain trajectory optimization with MPC.

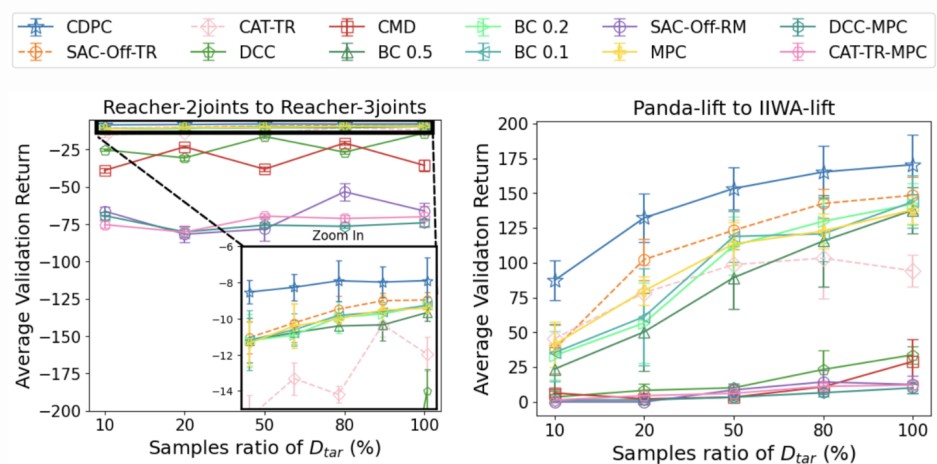

Figure 13: **Sample efficiency of CDPC and the benchmark methods:** CDPC demonstrates greater efficiency compared to the baseline methods across various dataset sizes, maintaining strong performance even as the dataset scale increases.

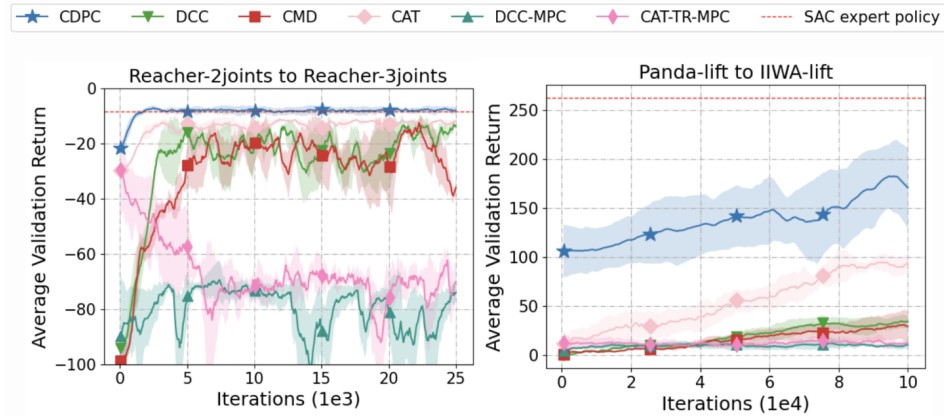

Figure 14: **Decoder performance of CDPC and the benchmark methods:** The learning curve of the CDPC decoder demonstrates a consistent improvement over the baseline methods, particularly in terms of convergence speed and final performance.

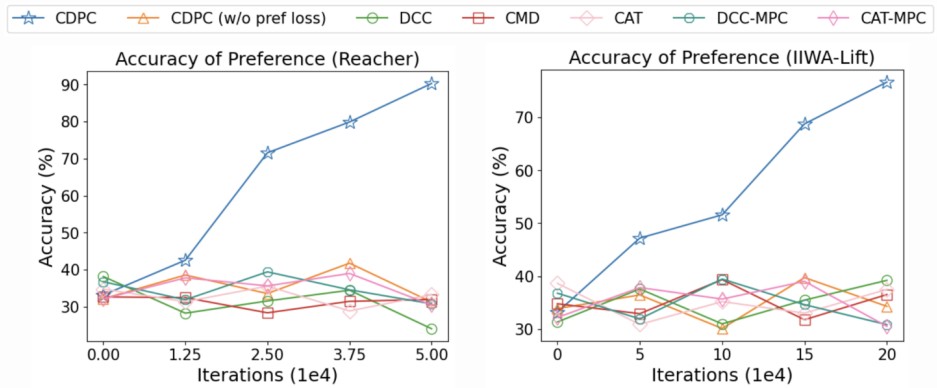

Figure 15: **Preference accuracy of the state decoders learned by CDPC, DCC, CMD, and CAT:** The integration of preference consistency loss enables CDPC to attain higher preference accuracy than the baseline methods.

### F.3 TRANSFER BETWEEN DIFFERENT TASKS ON THE SAME ROBOT

To further showcase the wide applicability of CDPC, we further evaluate CDPC on the transfer problems between different tasks within the same robotic environment. Specifically, we provide additional results on two pairs of robotic tasks:

- **MuJoCo**: Halfcheetah (source domain) and Halfcheetah-stand (target domain).
- **Robosuite**: Panda-BlockStacking (source domain) and Panda-PickAndPlace (target domain).

We report the experimental results on the sample efficiency, decoder performance, preference accuracy in Figure 16, Figure 17, and Figure 18, respectively. We can observe that CDPC can still successfully achieve cross-domain transfer between different tasks within the same robotic environment.

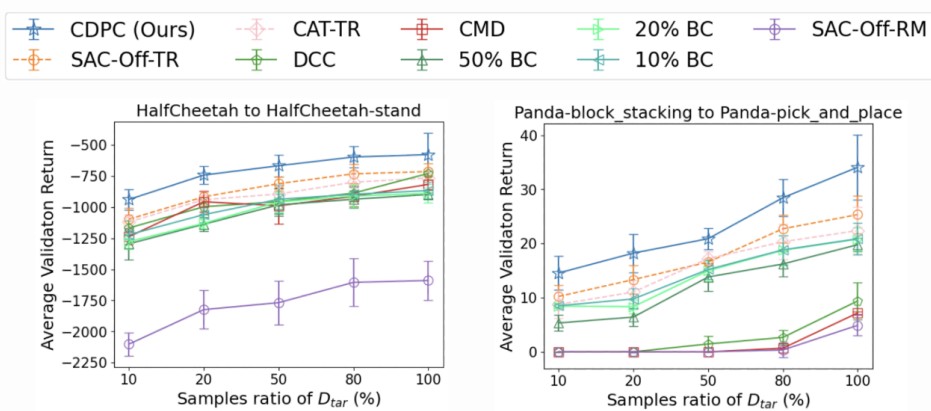

Figure 16: **Sample efficiency of CDPC and the benchmark methods:** CDPC demonstrates greater efficiency compared to the baseline methods across various dataset sizes, maintaining strong performance even as the dataset scale increases.

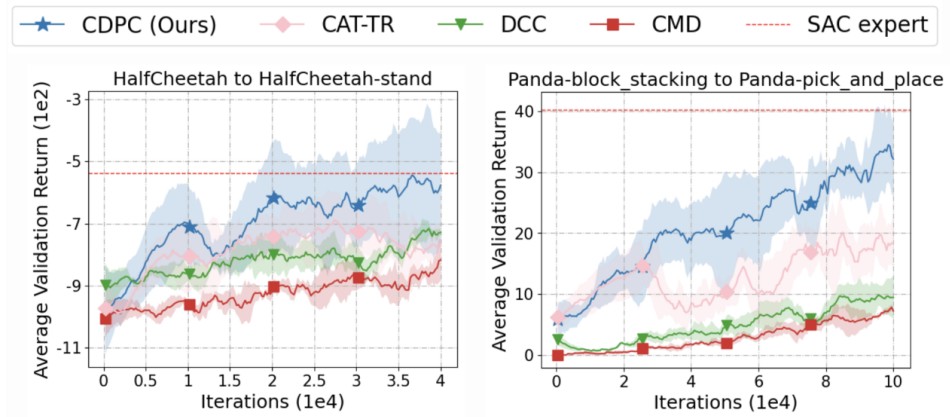

Figure 17: **Decoder performance of CDPC and the benchmark methods:** The learning curve of the CDPC decoder demonstrates a consistent improvement over the baseline methods, particularly in terms of convergence speed and final performance.

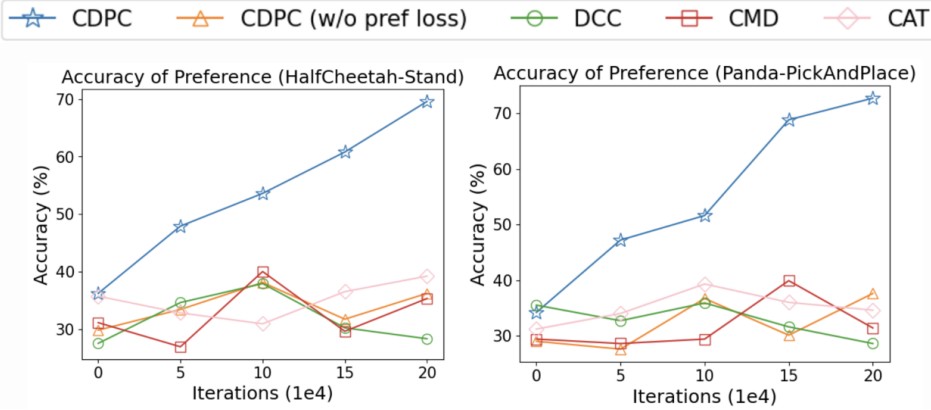

Figure 18: **Preference accuracy of the state decoders learned by CDPC, DCC, CMD, and CAT:** The integration of preference consistency loss enables CDPC to attain higher preference accuracy than the baseline methods.

