# OpenReview forum: "Cross-Domain Reinforcement Learning via Preference Consistency"
_ICLR.cc/2025/Conference — Submitted to ICLR 2025_

### Official Review · Reviewer_aVoh · 2024-10-21

**Soundness:** 2
**Presentation:** 3
**Contribution:** 2
**Rating:** 5
**Confidence:** 2

**Summary:**

This paper proposes a novel method, CDPC, to address the challenge of representation learning in cross-domain reinforcement learning. By leveraging trajectory preference signals from the target domain, the CDPC approach significantly enhances learning efficiency during knowledge transfer between the source and target domains.

**Strengths:**

1. Studies an important problem of cross-domain RL from offline data.

2. Shows a counterexample of the exist unsupervised CDRL issue (figure 1) and prpose a principle of cross-domain preference consistency to address this issue.

3. Clear writing in most places.

4. The experimental results on MuJoCo and RoboSuite demonstrate that our approach outperforms existing Cross-Domain Reinforcement Learning (CDRL) methods.

**Weaknesses:**

1. The reason for using MPC instead of other planning methods during the testing phase has not been clearly explained.

2. Although the experiments demonstrate that the combination of PBRL and MPC is highly effective in the context of CDRL, the advantages and underlying mechanisms of this integration have not been thoroughly analyzed. The work lacks theoretical exploration and a detailed discussion of its contributions.

3. The layout of the experimental results on page 10 requires revision.

4. Equations (3) and (5) should indeed be written with ":=" for consistency, just like Equations (2) and (4). Ensuring uniform formatting across all equations is important for clarity and professionalism in the presentation.

**Questions:**

1. How is "weak supervision" defined in the paper? No specific information has been provided. Please clarify this in the introduction section.

2. Are there any theoretical justification for the CDPC?

---

> ### Author Response · Authors · 2024-12-01
> **Response to Reviewer aVoh**
>
> We sincerely thank the reviewer for the insightful feedback. Below, we address the questions raised by the reviewer via the point-to-point response. We hope the response could help the reviewer further recognize our contributions. Thank you.
>
> **Q1: Provide a more detailed discussion of the contributions.**
>
> A1: We highlight the innovative aspects of this paper as follows:
>
> 1. **A new insight – correspondence identifiability issue, into the conventional unsupervised cross-domain RL**: We identify the correspondence identifiability issue in unsupervised cross-domain RL, which typically relies on the dynamics cycle consistency loss (e.g., the DCC and CMD algorithms) to learn the state-action correspondence. This issue indicates that without any supervision from the target domain, learning the state-action correspondence can be an underdetermined problem. Moreover, this insight motivates the research question on how to address this issue with only weak supervision signals.
>
> 2. **A new cross-domain RL formulation – Cross-Domain Preference-based RL (CD-PbRL)**: Inspired by the classic preference-based RL (PbRL) and the recent works on the fine-tuning of language models, we propose the CD-PbRL problem, where the learner receive a small amount of pairwise preference feedback from the target domain to tackle the above identifiability issue.
>
> 3. **A new cross-domain RL concept – Cross-Domain Preference Consistency (CDPC)**: Our core insight is that pairwise preference implicitly encodes the underlying goal of the task, and hence the consistency in preference across the source and target domains indicates their domain similarity. This motivates the CDPC framework, which can better learn the state-action correspondence by enforcing the trajectory preferences to be aligned across the two domains, based on the intuition that a policy is transferable across domains if the source and target domains have better consensus on the preference over trajectories under a learned inter-domain mapping.
>
> 4. **A practical implementation of CDPC**: To substantiate the CDPC framework, we propose two novel modules: (i) Cross-domain preference loss, which helps us effectively learn a target-to-source state decoder by ensuring preference consistency across domains. (ii) Cross-domain model predictive control for inference: During inference, we propose to leverage the learned state decoder and determine the target-domain actions by planning via cross-domain model-predictive control (MPC). This design leads to a simpler implementation as there is no need to learn the action correspondence between source and target domains.
>
> In summary, this paper systematically addresses cross-domain RL through novel insights, a new formulation, and a new algorithm with practical implementation.
>
>
> **Q2: Explain the reason for using MPC instead of other planning methods during the testing phase.**
>
> A2: Thank you for raising this point. As described in the Section 1 and Section 5.1 of the paper, the design of **selecting target-domain actions with cross-domain trajectory optimization** is very general and can be implemented by MPC or any off-the-shelf planning method. In this paper, we choose MPC as an exemplar to showcase the effectiveness for the following reasons:
>
> (1) MPC, also known as receding horizon control, is known to be more adaptive in the sense that it can replan and refit the sampling distribution at each step. Moreover, MPC is also known to handle constraints on the state and action space more easily (Kamthe and Deisenroth, 2017), and hence it is especially popular in robot control tasks.
>
> (2) In the context of RL, MPC has recently been widely adopted and shown to be quite competitive in various locomotion and robot arm manipulation tasks [Argenson and Dulac-Arnold, 2021; Hansen et al., 2022; Hansen et al., 2024]. Therefore, it appears quite natural to use MPC in our CDPC.
>
> **References:**
>
> [Hansen et al., 2022] Nicklas Hansen, Xiaolong Wang, and Hao Su, “Temporal Difference Learning for Model Predictive Control," ICML 2022.
>
> [Hansen et al., 2024] Nicklas Hansen, Hao Su, and Xiaolong Wang, "TD-MPC2: Scalable, Robust World Models for Continuous Control," ICLR 2024.
>
> [Argenson and Dulac-Arnold, 2021] Arthur Argenson and Gabriel Dulac-Arnold, “Model-based offline planning,” ICLR 2021.

---

> ### Author Response · Authors · 2024-12-01
> **Response to Reviewer aVoh**
>
> **Q3: How is "weak supervision" defined in the paper? No specific information has been provided. Please clarify this in the introduction section.**
>
> A3: In the context of RL, a weakly-supervised setting refers to scenarios where the learners rely on indirect supervision, such as human preferences or rankings, rather than explicit reward labels, to learn well-performing policies [Lee et al., 2020; Wang et al., 2022]. We have added this in the updated manuscript.
>
> **References:**
>
> [Lee et al., 2020] Lisa Lee, Ben Eysenbach, Russ R Salakhutdinov, Shixiang Shane Gu, and Chelsea Finn, “Weakly-supervised reinforcement learning for controllable behavior,” NeurIPS 2020
>
> [Wang et al., 2022] Zihan Wang, Zhangjie Cao, Yilun Hao, and Dorsa Sadigh, “Weakly supervised correspondence learning,” ICRA 2022.
>
> **Q4: Are there any theoretical justification for the CDPC?**
>
> A4: Thank you for the helpful question.
>
> Indeed, CDPC can be theoretically justified by connecting the theory of PbRL with cross-domain RL:
>
> (1) Notably, in the PbRL literature, it has been theoretically shown (e.g., by [Xu et al., 2020]) that the preferences over trajectories uniquely determine an optimal policy (in the sense of reward maximization) under some mild conditions (e.g., Assumption 1 in [Xu et al., 2020]).
>
> Moreover, [Xu et al., 2020] also shows that with probability at least $1-\delta$, an $\epsilon$-optimal policy can be learned from $O(poly(H,S,A) \log(1/\delta) / \epsilon)$ pairwise comparisons. This suggests that a near-optimal policy can be obtained by observing only finitely pairwise preferences.
>
> (2) Inspired by the above theoretical result, we adapt this idea to the cross-domain RL setting. Specifically, under a state decoder $\phi^{-1}$, if one can have perfect cross-domain preference consistency (i.e., for any pair of target-domain trajectories, we have $\tau_{i} \prec \tau_{j}$ leads to $\phi^{-1}(\tau_{i}) \prec \phi^{-1}(\tau_{j})$, then this implies that an optimal policy in the source domain $\mathcal{M}\_{src}$ is also optimal for the target domain $\mathcal{M}\_{tar}$ under $\phi^{-1}$.
>
> More generally, if most of the pairwise trajectory preferences are consistent under some $\phi^{-1}$ across the two domains, then an optimal policy in the source domain $\mathcal{M}\_{src}$ shall also be nearly optimal for the target domain $\mathcal{M}_{tar}$.
>
> (3) From our experiments (cf., Figure 8), we observe that a high preference accuracy can indeed be achieved by the proposed CDPC loss (cf. Equation (2)), and this explains the favorable cross-domain transfer performance of the proposed CDPC method.
>
> **Reference:**
>
> [Xu et al., 2020] Yichong Xu, Ruosong Wang, Lin F. Yang, Aarti Singh, Artur Dubrawski, “Preference-based Reinforcement Learning with Finite-Time Guarantees,” NeurIPS 2020.
>
> **Q5: Equations (3) and (5) should indeed be written with ":=" for consistency, just like Equations (2) and (4). Ensuring uniform formatting across all equations is important for clarity and professionalism in the presentation.**
>
> A5: Thank you for catching this. We have fixed them in the updated manuscript.

---

### Official Review · Reviewer_i9LB · 2024-11-03

**Soundness:** 2
**Presentation:** 2
**Contribution:** 2
**Rating:** 3
**Confidence:** 4

**Summary:**

This paper introduces a novel framework for cross-domain reinforcement learning that leverages pairwise trajectory preferences in the target domain as weak supervision. The proposed Cross-Domain Preference Consistency (CDPC) principle aims to address the correspondence identifiability issue (CII) in unsupervised CDRL by aligning trajectory preferences across source and target domains. Extensive experiments in MuJoCo and Robosuite demonstrating the effectiveness of CDPC in knowledge transfer across domains.

**Strengths:**

-  The idea of cross-domain preference consistency is Interesting.
- The experimental results are comprehensive.

**Weaknesses:**

- In the standard unsupervised Cross-Domain Reinforcement Learning setting, the learner is typically provided with a set of target-domain trajectories that contain only state-action pairs, without any reward signal. The authors, however, introduce an additional preference signal, which the reviewer believes contravenes the foundational assumption of unsupervised learning. The introduction of a sufficient number of preference signals could potentially allow for the inference of the underlying reward function, thus undermining the unsupervised nature of the task. Furthermore, the reviewer notes a lack of specificity in the experimental section regarding the quantity of preference signals used. The manuscript does not clarify how many preference data points were utilized, what proportion of the dataset they represent, or how the performance of the model scales with the amount of preference data. It is crucial to understand the sensitivity of the model's performance to the quantity of preference data, as this could significantly impact the generalizability and robustness of the proposed method.

- Regarding Figure 1, a lack of clarity on the implications of the correspondence identifiability issue. It appears that both $\tau_\alpha$ and $\tau_\beta$ are optimal trajectories, and the existence of multiple optimal trajectories within the same environment is a common occurrence. The reviewer questions whether the authors aim to identify a unique one-to-one mapping between the source and target domains. The necessity for uniqueness is not immediately apparent, and the manuscript would benefit from a more detailed explanation of why a unique mapping is required and how it affects the overall goal of the cross-domain transferability of policies.

- Concerning Equation 3, the reviewer inquires about the origin of the transition kernel T of the source domain. It is unclear whether this is learned through training or directly accessible through an environmental model.

**Questions:**

See Weaknesses.

---

> ### Author Response · Authors · 2024-12-01
> **Response to Reviewer i9LB**
>
> We greatly appreciate the reviewer’s insightful feedback for improving our paper. Below, we address the questions raised by the reviewer by providing the point-to-point response. We hope the response could help the reviewer further recognize our contributions. Thank you.
>
> **Q1: In the standard unsupervised Cross-Domain Reinforcement Learning setting, the learner is typically provided with a set of target-domain trajectories that contain only state-action pairs, without any reward signal. The authors, however, introduce an additional preference signal, which the reviewer believes contravenes the foundational assumption of unsupervised learning. The introduction of a sufficient number of preference signals could potentially allow for the inference of the underlying reward function, thus undermining the unsupervised nature of the task.**
>
> A1: Thank you for your question. We would like to first clarify the possible misunderstanding regarding our work. **One main contribution of this paper is to identify the fundamental issue of unsupervised CDRL and accordingly advocate for weakly-supervised CDRL. Specifically**:
>
> 1. **Correspondence identifiability issue (CII) can occur under unsupervised cross-domain RL methods**: As mentioned by the reviewer, unsupervised cross-domain RL methods (such as DCC [Zhang et al., 2021] and CMD [Gui et al., 2023]) typically learn the state-action correspondence (i.e., a target-to-source state decoder and a source-to-target action encoder) by minimizing a dynamics cycle consistency loss $L_{dcc}$, which aligns the one-step transition of the unpaired trajectories from the two domains. However, we show that $L_{dcc}=0$ does not necessarily lead to a good state decoder $\phi$ through an example where multiple decoders with $L_{dcc}=0$ but different resulting total return can indeed exist (cf. Figure 1). This verifies the existence of the correspondence identifiability issue and why the unsupervised setting can be problematic and hence not preferred.
> Please see our response to Q3 below for the detailed description of this example.
>
> 2. **CD-PbRL, a new cross-domain RL formulation with preference feedback as weak supervision signal**: Given that the fully unsupervised setting is not preferred due to CII, we address this by proposing a new formulation – CD-PbRL, where the agent in the target domain can receive additional weak supervision signal in the form of preferences over trajectory. Such pairwise preference feedback has been considered in the single-domain RL. In our work, one of our main contributions is to show that pairwise preference feedback can also help in cross-domain RL.
>
> 3. **The proposed CDPC algorithm indeed achieves effective cross-domain transfer and can perform better than the baselines with either the true reward function (i.e., SAC-Off-TR, CAT-TR) or a reward model learned from preference feedback (i.e., SAC-Off-RM)**: As mentioned by the reviewer, we agree that preference signals can indeed be used for the inference of the underlying reward function. Despite this, through extensive experiments, we observe that CDPC can nicely transfer the knowledge from the source domain to the target domain based only on preference feedback, and hence outperform those baselines with either the true reward function (i.e., SAC-Off-TR, CAT-TR) or a learned reward function (i.e., SAC-Off-RM).
>
> In summary, our paper demonstrates that CD-PbRL is a promising new CDRL setup, and this can be solved effectively by the proposed CDPC algorithm.

---

> ### Author Response · Authors · 2024-12-01
> **Response to Reviewer i9LB**
>
> **Q2: The manuscript does not clarify how many preference data points were utilized, what proportion of the dataset they represent, or how the performance of the model scales with the amount of preference data. It is crucial to understand the sensitivity of the model's performance to the quantity of preference data, as this could significantly impact the generalizability and robustness of the proposed method.**
>
> A2:
>
> **(1) Size of target-domain datasets**: Regarding the size of target-domain data, the size of $\mathcal{D}\_{tar}$ is $5\times 10^5$ transition pairs, which corresponds to 500 trajectories for HalfCheetah and Walker, and 1000 trajectories for Robosuite tasks. Notably, compared to the experimental configurations of the baselines in their original papers and other single-domain preference-based RL papers, the amount of data we use is either comparable or relatively small. For example:
> - In D4RL [Hansen et al., 2023], both SAC-Off-TR and BC used 1 million transition pairs.
> - DCC [Zhang et al., 2021] used 50,000 trajectories.
> - CAT-TR [You et al., 2022], trained in an online setting, used a substantial amount of data up to 3 million transition pairs.
> - Other single-domain preference-based RL papers: For example, Uni-RLHF [Yuan et al., 2024] used up to 1,000 trajectories. [Kang et al., 2023] leverages the D4RL dataset and also used 1000 trajectories with 50,000 preference feedback queries by default.
>
> Recall from Figure 5 of the original manuscript that we evaluate CDPC and other benchmark methods under different sizes of the target-domain dataset. Notably, CDPC can already achieve a moderately high total return on all the six tasks by only using as few as 10% of the target-domain dataset, which corresponds to only 50 trajectories for HalfCheetah and Walker, and 100 trajectories for Robosuite tasks.
>
> **(2) Quantify of preference data (i.e., number of preference queries):**
>
> Recall from Figure 5 of the original manuscript that we evaluate CDPC and other benchmark methods under different sizes of the target-domain dataset, i.e., $10\%, 20\%, 50\%, 80\%, 100\%$. The corresponding numbers of preference queries $N_{q}$ are summarized as follows:
>
> - For the results under $10%$ of $\mathcal{D}\_{tar}$:
>     - Halfcheetah and Hopper: $N\_{q}=1250$
>     - Robosuite: $N\_{q}=2500$
>     - Reacher: $N\_{q}=16000$
>
> - For the results under $20%$ of $\mathcal{D}\_{tar}$:
>     - Halfcheetah and Hopper: $N\_{q}=2500$
>     - Robosuite: $N\_{q}=5000$
>     - Reacher: $N\_{q}=16000$
>
> - For the results under $50%, 80\%, 100\%$ of $\mathcal{D}\_{tar}$:
>     - Halfcheetah and Hopper: $N\_{q}=6400$
>     - Robosuite: $N\_{q}=6400$
>     - Reacher: $N\_{q}=16000$
>
> Notably, the above numbers are comparable to other single-domain PbRL works. For example, [Kang et al., 2023] uses 50,000 queries by default and PEBBLE [Lee et al., 2021] uses from 2500 to 50,000 queries in their experiments on robot arm manipulation.
>
> To highlight that CDPC indeed can learned with only a fairly reasonable number of preference labels and a fairly small target-domain dataset, we summarize the testing reward performance in the case of $10%$ of $\mathcal{D}_{tar}$:
>
>
> |           | Halfcheetah      | Hopper       | Reacher       | IIWA-Lift       |IIWA-Door       | IIWA-NutAssemblyRound |
> |---------------------|------------------|--------------------|--------------------|--------------------|--------------------|--------------------|
> | CDPC (Ours) | **1961.78$\pm$165.23** | **585.22$\pm$ 93.42** |  **-8.53$\pm$0.65** | **87.32$\pm$14.29** |  **284.56$\pm$22.15**       |   **23.01$\pm$2.49**      |
> | SAC-Off-TR          | 1010.45$\pm$178.32 | 343.34$\pm$90.01 |   -11.05$\pm$0.87  |  38.01$\pm$17.56       |  230.23$\pm$15.89       | 39.11$\pm$3.34        |
> | SAC-Off-RM          | -435.55$\pm$105.54 | -301.24$\pm$53.97 |   -66.31$\pm$2.67      |  0.00$\pm$0 .00      |  0.00$\pm$0.00       |   0.00$\pm$0.00      |
> | CAT-TR       | -121.56$\pm$185.09        |   300.22$\pm$91.38      |  -15.46$\pm$1.24       |   45.44$\pm$10.56      |  86.75$\pm$14.58       |   10.89$\pm$4.56      |
> | DCC          |  -33.33$\pm$144.79       |  232.67$\pm$108.04       |   -25.21$\pm$0.66      |  3.53$\pm$2.28       |  0.89$\pm$0.10       |  0.00$\pm$0.00       |
> | CMD          |  -341.65$\pm$322.61       |   223.54$\pm$79.62      |   -38.99$\pm$1.64      | 6.45$\pm$2.56        |  0.00$\pm$0.00       |   0.31$\pm$0.10      |
> | 10\% BC        | 458.11$\pm$807.33        |  298.99$\pm$33.57       |  -11.19$\pm$1.66       |  35.45$\pm$20.45       |  150.50$\pm$11.34       |  13.23$\pm$3.34       |
> | 20\% BC        |  789.56$\pm$466.13       |  253.34$\pm$57.66       |   -11.22$\pm$1.45      |  33.33$\pm$17.56       |  134.45$\pm$30.45       |   13.22$\pm$5.78      |
> | 50\% BC        | 801.21$\pm$182.28        |   215.22$\pm$125.47      |   -11.21$\pm$1.24      |  23.66$\pm$15.67       |  102.33$\pm$12.45       |    12.89$\pm$1.34     |
>
> (References are in the response below)

---

> ### Author Response · Authors · 2024-12-01
> **Response to Reviewer i9LB**
>
> **References:**
>
> [Zhang et al., 2021] Qiang Zhang, Tete Xiao, Alexei A Efros, Lerrel Pinto, and Xiaolong Wang, "Learning cross-domain correspondence for control with dynamics cycle-consistency," ICLR 2021.
>
> [Gui et al., 2023] Haiyuan Gui, Shanchen Pang, Shihang Yu, Sibo Qiao, Yufeng Qi, Xiao He, Min Wang,and Xue Zhai, "Cross-domain policy adaptation with dynamics alignment," Neural Networks, 2023.
>
> [You et al., 2022] Heng You, Tianpei Yang, Yan Zheng, Jianye Hao, and Matthew E. Taylor, "Cross-domain adaptive transfer reinforcement learning based on state-action correspondence," UAI 2022.
>
> [Yuan et al., 2024] Yifu Yuan, Jianye Hao, Yi Ma, Zibin Dong, Hebin Liang, Jinyi Liu, Zhixin Feng, Kai Zhao, and Yan Zheng, "Uni-RLHF: Universal Platform and Benchmark Suite for Reinforcement Learning with Diverse Human Feedback," ICLR 2024.
>
> [Kang et al., 2023] Yachen Kang, Diyuan Shi, Jinxin Liu, Li He, Donglin Wang, “Beyond Reward: Offline Preference-guided Policy Optimization,” ICML 2023
>
> [Lee et al., 2021] Kimin Lee, Laura Smith, Pieter Abbeel, “PEBBLE: Feedback-Efficient Interactive Reinforcement Learning via Relabeling Experience and Unsupervised Pre-training,” ICML 2021.
>
>
> **Q3: Regarding Figure 1, a lack of clarity on the implications of the correspondence identifiability issue. It appears that both $\tau\_{\alpha}$ and $\tau\_{\beta}$ are optimal trajectories, and the existence of multiple optimal trajectories within the same environment is a common occurrence. The reviewer questions whether the authors aim to identify a unique one-to-one mapping between the source and target domains. The necessity for uniqueness is not immediately apparent, and the manuscript would benefit from a more detailed explanation of why a unique mapping is required and how it affects the overall goal of the cross-domain transferability of policies.**
>
> A3: Thank you for the helpful suggestion. To make the correspondence identifiability issue and the need for a unique mapping more transparent, we slightly augment the gridworld example as follow:
>
> An illustration of the example is available at https://imgur.com/a/9SdiUxO (also in Figure 1 of the updated manuscript)
>
> **Problem setup:** Consider one target domain trajectory $\tau$, two state decoder $\phi\_\alpha^{-1}$ and $\phi\_\beta^{-1}$, one well-trained source domain policy $\pi\_{src}$, source domain reward function $R\_{src}$, which is defined as follows: the top-left corner is the starting point, the bottom-left corner contains the treasure, which provides a reward of +0.5 upon reaching it, and the bottom-right corner is the goal, which provides a reward of +1 and terminates the episode. For simplicity, let us assume the discount factor $\gamma$ equals 1.
>
>
> (1) For $\phi\_\alpha^{-1}$, the process of decoding can be described as follows:
> - $\phi\_\alpha^{-1}(00, 00)\Rightarrow(0,0)$, $\pi\_{src}(0,0)=\rightarrow$, go to $(0,1)$, reward = +0
> - $\phi\_\alpha^{-1}(00, 01)\Rightarrow(0,1)$, $\pi\_{src}(0,1)=\rightarrow$, go to $(0,2)$, reward = +0
> - $\phi\_\alpha^{-1}(00, 10)\Rightarrow(0,2)$, $\pi\_{src}(0,2)=\downarrow$, go to $(1,2)$, reward = +0
> - $\phi\_\alpha^{-1}(01, 10)\Rightarrow(1,2)$, $\pi\_{src}(1,2)=\downarrow$, go to $(2,2)$, reward = +1
> - $\phi\_\alpha^{-1}(10, 10)\Rightarrow(2,2)$, total return = 1
>
>
> (2) For $\phi_\beta^{-1}$, the process of decoding can be described as follows:
> - $\phi\_\beta^{-1}(00, 00)\Rightarrow(0,0)$, $\pi\_{src}(0,0)=\downarrow$, go to $(1,0)$, reward = +0
> - $\phi\_\beta^{-1}(00, 01)\Rightarrow(1,0)$, $\pi\_{src}(1,0)=\downarrow$, go to $(2,0)$, reward = +0.5
> - $\phi\_\beta^{-1}(00, 10)\Rightarrow(2,0)$, $\pi\_{src}(2,0)=\Rightarrow$, go to $(2,1)$, reward = +0
> - $\phi\_\beta^{-1}(01, 10)\Rightarrow(2,1)$, $\pi\_{src}(2,1)=\rightarrow$, go to $(2,2)$, reward = +1
> - $\phi\_\beta^{-1}(10, 10)\Rightarrow(2,2)$, total return = 1.5
>
> Based on the above, we cannot determine whether $\tau\_\alpha'$ or $\tau\_\beta'$ is better, without considering total return. As a result, it remains infeasible to distinguish between them if we only use dynamic cycle consistency loss. Without a suitable mechanism for choosing between $\phi\_\alpha^{-1}$ or $\phi\_\beta^{-1}$, the correspondence identifiability issue could easily arise.

---

> ### Author Response · Authors · 2024-12-01
> **Response to Reviewer i9LB**
>
> **Q4: Concerning Equation 3, the reviewer inquires about the origin of the transition kernel $T$ of the source domain. It is unclear whether this is learned through training or directly accessible through an environmental model.**
>
> A4: For the reviewer’s convenience, we restate the CD-PbRL formulation (cf. Section 4) as follows:
> - There are two domains, the source domain $\mathcal{M}\_{src}$ and the target domain $\mathcal{M}\_{tar}$.
> - Regarding the source domain, for efficient knowledge transfer, the source domain is typically an environment that is cheap and easy to access, e.g., a simulator. Therefore, we presume that the learner can have full access to the source environment (including the transition kernel). This is the conventional setting adopted by many CDRL works, such as DCC and CMD.
> - By contrast, in the target domain, we are only given a dataset $\mathcal{D}\_{tar}$, which contains multiple trajectories composed of states and preferences between pairs of trajectories.
>
> Based on the above CD-PbRL setup (cf. Section 4), the transition kernel $\mathcal{T}$ of the source domain is directly accessible.

---

### Official Review · Reviewer_DetX · 2024-11-04

**Soundness:** 3
**Presentation:** 3
**Contribution:** 3
**Rating:** 6
**Confidence:** 4

**Summary:**

For cross-domain transfer in RL, the paper proposes a CDPC method using preference consistency between the target and source domains. The proposed approach learns a state decoder to connect target domain trajectories with source domain trajectories, and then use MPC to select actions for the target domain based on the corresponding source domain rewards. Experiments show data efficiency of the proposed CDPC method and improved final performance compared to some prior methods.

**Strengths:**

- The ability to transfer knowledge across two RL domains is important for data efficiency but also challenging. This paper proposes a method based on the idea of preference consistency when we only have preferences over trajectories in the target domain. The idea is to find a mapping such that the preference ordering of trajectories in the source domain and the corresponding trajectories in target domain are consistent. Then based on this mapping, one can get the preference of any two target domain trajectories by mapping them back into the source domain and utilizing our source domain knowledge, which then allows one to apply MPC to select the best action. The idea is pretty novel and seems to be effective.

-  Multiple experiments are conducted to evaluate different aspects of the proposed CDPC method. It is shown that CDPC performs better than prior methods in several domain transfer problems, and ablation studies show the importance of the proposed preference consistency loss in terms of preference accuracy and final performance.

**Weaknesses:**

- In addition to the learned decoder, CDTO requires to learn a target-domain dynamics model, but little detail is provided for the target-domain dynamics model. Are the samples used in learning the target-domain dynamics taken into account in the sample efficiency calculation compared with other methods? Are there experiments showing the importance of the quality of this learned dynamics model?

**Questions:**

- For the preference accuracy in Figure 8, how is the preference accuracy evaluated? Are they evaluated on the same set of trajectories for all the methods?

---

> ### Author Response · Authors · 2024-12-01
> **Response to Reviewer DetX**
>
> We greatly appreciate the reviewer’s insightful feedback for improving our paper. We provide our point-by-point response as follows. We hope the response could help the reviewer further recognize our contributions. Thank you.
>
> **Q1: In addition to the learned decoder, CDTO requires learning a target-domain dynamics model, but little detail is provided for the target-domain dynamics model. Are the samples used in learning the target-domain dynamics taken into account in the sample efficiency calculation compared with other methods? Are there experiments showing the importance of the quality of this learned dynamics model?**
>
> A1: Thank you for the helpful questions. Yes, our target-domain dynamics model was trained based on the same dataset as the decoder $\phi^{-1}$, and therefore no additional data was used.
>
> To evaluate the robustness of CDPC to the quality of the learned dynamics model, we conduct an additional empirical study by adding additional perturbation noise to the predicted states output by the dynamics model. Intuitively, one shall expect that the decision made by the MPC procedure can be affected by the perturbation noise. Specifically:
> - We first generate Gaussian random variables with zero mean and a standard deviation of $\alpha$, which controls the perturbation strength.
> - To ensure a unified comparison, based on the state representations provided by the official MuJoCo and Robosuite documentation, the noise terms are further rescaled according to the range of each dimension of the state. In this way, $\alpha$ directly reflects the relative noise magnitude compared to the range of states.
>
> The experimental results are provided in the table below.
>
> |           | Reacher          | IIWA-Lift          |
> |---------------------|------------------|--------------------|
> | CDPC ($\alpha=0.0$) | -7.9 $\pm$ 1.29  | 170.45 $\pm$ 21.49 |
> | CDPC ($\alpha=0.1$) | -8.05 $\pm$ 1.32 | 166.23 $\pm$ 20.09 |
> | CDPC ($\alpha=0.2$) | -8.31 $\pm$ 1.21 | 162.01 $\pm$ 22.06 |
> | CDPC ($\alpha=0.4$) | -8.82 $\pm$ 1.57 | 158.67 $\pm$ 18.35 |
> | SAC-Off-RM          | -66.23 $\pm$ 5.55 | 12.56 $\pm$ 6.09 |
> | SAC-Off-TR          | -8.97 $\pm$ 0.43 | 148.44 $\pm$ 13.24 |
> | CAT-TR          | -11.97 $\pm$ 0.94 | 94.20 $\pm$ 11.27 |
> | DCC          | -14.02 $\pm$ 1.22 | 33.90 $\pm$ 6.28 |
> | CMD          | -35.68 $\pm$ 3.9 | 29.13 $\pm$ 15.88 |
>
> We can observe that despite the lowered quality of the learned dynamics model, the performance of CDPC is only slightly affected and still remains fairly robust and superior to the strong benchmark SAC-Off-TR, which learns directly from the true target-domain reward function.
>
> **Q2: For the preference accuracy in Figure 8, how is the preference accuracy evaluated? Are they evaluated on the same set of trajectories for all the methods?**
>
> A2: We provide the detailed procedure of the evaluation of preference accuracy used by CDPC and other benchmark methods in Figure 8 as follows:
> - Step 1: Collect a target-domain dataset $\mathcal{D}_{\text{tar}}^\prime$ of trajectories with preference labels.
> - Step 2: Randomly sample a batch of $k$ trajectory pairs $\\{ (\tau_1^{(i)}, \tau_2^{(i)}) \\}_{i=1}^{k}$ and the corresponding preference labels $y^{(1)},\cdots,y^{(k)}$ from the target-domain dataset.
> - Step 3: Feed each pair $(\tau_1^{(i)}, \tau_2^{(i)})$ into the learned state decoder $\phi^{-1}$ and get the corresponding source-domain trajectories $(\tau_1^{(i)\prime}, \tau_2^{(i)\prime})$. Accordingly, let $z^{(i)}$ denote the source-domain preference label of $(\tau_1^{(i)\prime}, \tau_2^{(i)\prime})$.
> - Step 4: Compute $\text{Accuracy} = \frac{\sum_{i=1}^{k} \mathbb{I}\\{y^{(i)} = z^{(i)}\\}}{k} \times 100\\%$.
>
> Moreover, all the methods are evaluated based on the same set of trajectories in $\mathcal{D}_{\text{tar}}^\prime$.

---

### Official Review · Reviewer_pqN5 · 2024-11-04

**Soundness:** 3
**Presentation:** 3
**Contribution:** 2
**Rating:** 5
**Confidence:** 4

**Summary:**

The paper describes a new method of cross domain reinforcement learning
based on the idea of receiving target domain preference labels rather than
target domain reward labels.  It proposes training an encoder/decoder to
map between source and target states including previouly proposed
consistency and reconstruction losses and adding in a preference loss.  It
also adds MPC at inference time to improve the action selection.  It
demonstrates its results in simulation using some standard cross-domain
experiments and compares to several methods using cross-domain approaches
and some that don't.

**Strengths:**

The paper is well written and easy to understand and I believe it presents
a new idea.

**Weaknesses:**

I have a few major concerns which are listed here:

1. I did not find the problem setup very compelling.  Its hard to imagine
robotics applications where its easy to get preference data but the reward
function is unknown.  This might well happen in language models or other
similar applications.  But the empirical studies were done on robotics
examples where the reward functions are well known.  In order to be
convincing, I think the empirical studies should be done in applications
where this problem setup really happens.

2. Cross domain preference consistency seems to make sense in the
applications that were shown -- basically you change the
kinematics/dynamics of a robot that still has roughly the same task
capability and does the same task.  Maybe that really is your main use
case, but that again leads to point 1 above.  In robotics it would be nice
to be to change the task for the same robot, but it seems unlikely that
CDPC applies in that case.

3. The effects of MPC are not well tested in the empirical study.  It seems
possible that the main thing making the algorithm perform well is the MPC
rather than the decoder learned via eq 5.  A useful comparison would be to
take the same target domain dynamics model to generate trajectories and
evaluate those trajectories with the critic learned by SAC-Off-TR and by
SAC-Off-RM.  You similarly could test CAT-TR plus MPC and DCC plus MPC to
see whether MPC is useful for resolving some of their limitations.


Here are some additional minor notes:

1. I did not find figure 1 helpful.  I agree some version of this figure
would be appropriate.  I found the decimal and binary distinction to be
confusing and maybe unnecessary.  While it was clear there were two
decoders, it was less clear how/why they both had zero cycle consistency
loss.

2. SAC-Off-TR is advertised as a "topline that should be an upper bound".
I don't believe that statement is accurate since while it does have the
benefit of the true reward function, it does not have the benefit of the
source domain information.

3. The empirical results figures are too small to read and on several I had
to just go with the text and ignore the figure.  Figs 5, 6, and 11 could
probably be remedied by simply scaling up to use all the horizontal space
available.  The others need bigger reformatting.  This would take up more
vertical space, but you could deduplicate and tighten up some of the text
in the early sections to make room.

**Questions:**

Please comment on the concerns listed above, focusing on the first three listed as "major" concerns.

---

> ### Author Response · Authors · 2024-12-01
> **Response to Reviewer pqN5**
>
> We sincerely appreciate reviewer's insightful feedback for improving our paper. Below, we address the questions raised by the reviewer by providing the point-to-point response. We hope the response could help the reviewer further recognize our contributions. Thank you.
>
> **Q1: It’s hard to imagine robotics applications where it is easy to get preference data but the reward function is unknown. This might well happen in language models or other similar applications. But the empirical studies were done on robotics examples where the reward functions are well known. In order to be convincing, I think the empirical studies should be done in applications where this problem setup really happens.**
>
> A1: Thank you for raising this point. We would like to highlight that there are indeed many robotic applications where the reward function is either unknown or difficult to design and hence collecting preference feedback is preferred. For example:
>
> - **Teach robots to perform novel and complex behaviors**: As in one of the earliest work on preference-based RL [Christiano et al., 2017], one classic example is ``Hopper backflip,” where a hopper robot is trained to perform a sequence of backflips and end up landing upright. Under such a complex task, a proper reward function can be hard to specify while preference feedback can be easier to collect. Subsequently, similar examples has been adopted by many preference-based RL works, such as [Lee et al., 2021a; Yuan et al., 2024]
>
> - **Safe learning from human evaluative feedback**: As showcased by [Hiranaka et al., 2023], human evaluative preference feedback (“good”, “neutral”, or “bad”) can be useful in robotics as the humans can evaluate robot actions from the skill and parameter visualizations, without the need for watching the robot take action or human demonstrations (e.g., demos at https://seediros23.github.io/). As a result, human preference feedback can help achieve safe robot learning.
>
> - **Train real-world physical robots with real sensor inputs**: Real physical robots need to control the actuators based on the sensor input. If one would like to train physical robots to perform some common tasks in the human world (e.g., wash the dishes and scramble an egg), handcrafting a reward function can be challenging as the reward function needs to be a function of the robot’s sensors and actuators. By contrast, preference feedback will be easier to collect [Christiano et al., 2017; Wang et al., 2024].
>
> - **Improve the alignment of robot behavior with humans and avoid reward exploitation**: It is known that robots learned by RL can often discover ways to achieve high rewards via unexpected, unintended means. This is typically known as the misalignment or reward exploitation issue. To address this, learning from human preference feedback is one widely adopted approach [Lee et al., 2021b; Kim et al., 2023; Hejna and Sadigh, 2023]
>
>
> As a result of the above, there indeed exist several Preference-based RL benchmarks for robotics applications, such as B-Pref [Lee et al., 2021a] and Uni-RLHF [Yuan et al., 2024]. These further confirm that PbRL as well as our proposed cross-domain PbRL can be naturally applicable to the robotic domains.
>
> **References:**
>
> [Christiano et al., 2017] Paul Christiano, Jan Leike, Tom B. Brown, Miljan Martic, Shane Legg, Dario Amodei, “Deep reinforcement learning from human preferences,”
>
> [Lee et al., 2021a] Kimin Lee, Laura Smith, and Pieter Abbeel, “B-Pref: Benchmarking Preference-Based Reinforcement Learning,” NeurIPS 2021.
>
> [Yuan et al., 2024] Yifu Yuan, Jianye Hao, Yi Ma, Zibin Dong, Hebin Liang, Jinyi Liu, Zhixin Feng, Kai Zhao, and Yan Zheng, "Uni-RLHF: Universal Platform and Benchmark Suite for Reinforcement Learning with Diverse Human Feedback," ICLR 2024.
>
> [Hiranaka et al., 2023] Ayano Hiranaka, Minjune Hwang, Sharon Lee, Chen Wang, Li Fei-Fei, Jiajun Wu, and Ruohan Zhang, "Primitive Skill-based Robot Learning from Human Evaluative Feedback," IROS 2023.
>
> [Lee et al., 2021b] Kimin Lee, Laura Smith, Pieter Abbeel, “PEBBLE: Feedback-Efficient Interactive Reinforcement Learning via Relabeling Experience and Unsupervised Pre-training,” ICML 2021.
>
> [Kim et al., 2023] Changyeon Kim, Jongjin Park, Jinwoo Shin, Honglak Lee, Pieter Abbeel, and Kimin Lee, “Preference Transformer: Modeling Human Preferences using Transformers for RL,” ICML 2023.
>
> [Hejna and Sadigh, 2023] Joey Hejna, Dorsa Sadigh, “Inverse Preference Learning: Preference-based RL without a Reward Function,” NeurIPS 2023.
>
> [Wang et al., 2024] Yufei Wang, Zhanyi Sun, Jesse Zhang, Zhou Xian, Erdem Biyik, David Held, and Zackory Erickson, “RL-VLM-F: Reinforcement Learning from Vision Language Foundation Model Feedback,” ICML 2024.

---

> ### Author Response · Authors · 2024-12-01
> **Response to Reviewer pqN5**
>
> **Q2: In robotics it would be nice to change the task for the same robot, but it seems unlikely that CDPC applies in that case.**
>
> A2: Recall from Section 4 that CDPC is designed to solve the general CD-PbRL problem, which considers a pair of source and target domains denoted by
> - $\mathcal{M}\_{src}=(\mathcal{S}\_{src}, \mathcal{A}\_\{src\}, \mathcal{T}\_{src}, R\_{src}, \mu\_{src}, \gamma)$
> - $\mathcal{M}\_{tar}=(\mathcal{S}\_{tar}, \mathcal{A}\_{tar}, \mathcal{T}\_{tar}, R\_{tar}, \mu\_{tar}, \gamma)$
>
> By “*change the task for the same robot*,” this means that $\mathcal{S}\_{src}=\mathcal{S}\_{tar}$ and $\mathcal{A}\_{src}=\mathcal{A}\_{tar}$, while the transition kernels and the reward functions can be different. As this setting is a special case of the general CD-PbRL, one can expect that CDPC still applies in this case.
>
> To corroborate this, we further evaluate CDPC on the transfer problems between different tasks within the same robotic environment. Specifically, we provide additional results on two pairs of robotic tasks (an illustrative image is available at https://imgur.com/YzI61EU):
> - **Locomotion**: Halfcheetah (source domain) and Halfcheetah-Stand (target domain). Note that the HalfCheetah-Stand task is originally designed by [Hua et al., 2023] and available at https://github.com/piao-0429/EAR/tree/master.
> - **Robot arm manipulation**: Panda-BlockStacking (source domain) and Panda-PickAndPlace (target domain).
>
> [Hua et al., 2023] Pu Hua, Yubei Chen, and Huazhe Xu, "Simple Emergent Action Representations from Multi-Task Policy Training," ICLR 2023.
>
> The results are available as follows:
> - **Sample efficiency**: https://imgur.com/a/j0lVrYr
> - **Decoder performance of cross-domain methods**: https://imgur.com/a/T9sebYA
> - **Preference accuracy of cross-domain methods**: https://imgur.com/a/bkR6cBI
>
> We can observe that CDPC can still successfully achieve cross-domain transfer between different tasks within the same robotic environment.
>
> **Q3: The effects of MPC are not well tested in the empirical study. It seems possible that the main thing making the algorithm perform well is the MPC rather than the decoder learned via Eq. (5). A useful comparison would be to take the same target domain dynamics model to generate trajectories and evaluate those trajectories with the critic learned by SAC-Off-TR and by SAC-Off-RM. You similarly could test CAT-TR plus MPC and DCC plus MPC to see whether MPC is useful for resolving some of their limitations.**
>
> A3: Thank you for the helpful suggestion. We further demonstrate that the empirical strength of the CDPC algorithm indeed mainly comes from the design of cross-domain preference consistency. To address this, we further compare CDPC with the three additional baselines as follows:
>
> - **MPC**: This method employs MPC directly in the target domain, without using transfer learning. Here, we use the same dynamics model for both the pure MPC method and CDPC. The purpose of including this baseline is to verify whether CDPC performs well simply because MPC itself is inherently strong.
>
> - **CAT-TR-MPC**: We integrate the state decoder learned by CAT-TR with our proposed cross-domain MPC to select the target-domain actions, similar to CDPC. Here, the main purpose is to verify whether the integration of cross-domain MPC and other cross-domain RL methods (like CAT) already achieves strong empirical performance.
>
> - **DCC-MPC**: Similar to CAT-TR-MPC, we further integrate the state decoder learned by DCC with our proposed cross-domain MPC subroutine for target-domain action selection. We call this new variant DCC-MPC. Again, the main purpose here is to check whether the integration of MPC and other cross-domain methods like DCC already achieves good empirical performance.
>
> The results are available at:
> - **Sample efficiency**: https://imgur.com/a/gvtco8o
> - **Decoder performance of cross-domain methods**: https://imgur.com/a/DunwsLW
> - **Preference accuracy of cross-domain methods**: https://imgur.com/a/MBQ1d8N
>
> We can observe the following facts:
> - **CDPC is indeed more sample-efficient than pure target-domain MPC**: CDPC still remains the best after the three MPC-based baselines are included. Notably, using MPC directly in the target domain can produce decent actions, resulting in a moderately high total return. However, pure target-domain MPC still underperforms CDPC since CDPC, as a cross-domain transfer method, nicely leverages the learned model from the source domain.
> - **CAT-TR-MPC and DCC-MPC suffer from low preference accuracy and hence do not perform well**: On the other hand, CAT-TR-MPC and DCC-MPC still suffer from ineffective transfer. This is because the state decoders of these methods are still not able to produce correct trajectory rankings even under the integration with the cross-domain MPC module. This issue is particularly evident from the results on preference accuracy (cf. https://imgur.com/a/MBQ1d8N).

---

> ### Author Response · Authors · 2024-12-01
> **Response to Reviewer pqN5**
>
> **Q4: SAC-Off-TR is advertised as a "topline that should be an upper bound". I don't believe that statement is accurate since while it does have the benefit of the true reward function, it does not have the benefit of the source domain information.**
>
> A4: Thank you for the helpful suggestion. To avoid the possible confusion, we modify the description about SAC-Off-TR in Section 6.1 as follows:
> *By leveraging the true target-domain environmental rewards, it serves as a natural and expectedly strong benchmark method, even without transfer learning.*

---

### Meta-Review · Area_Chair_XzV4 · 2024-12-19

**Metareview:**

Summary: This paper explores cross-domain reinforcement learning (RL) in the absence of reward signals in the target domain. Instead, it considers a novel setting where trajectory-level preferences are accessible in the target domain. The authors propose the CDPC method, which employs a domain consistency approach to leverage preference information in the target domain using a learned state encoder. Experimental results indicate that the proposed CDPC method performs well across multiple benchmarks.

Strengths:

- The problem setting of cross-domain RL is inherently interesting and impactful.
- The proposed CDPC method demonstrates better performance compared to prior approaches in several domain transfer scenarios.

Weaknesses:

- A significant issue, also highlighted by some other reviewers, is the lack of a clear and precise statement of the studied problem setting. Since the setting—where the target domain only provides preference information—is novel, a more thorough exploration of the quality and quantity of preference labels is essential.
- As this paper focuses on cross-domain RL, a more detailed analysis of domain differences is essential. While this concern has not been explicitly raised by other reviewers, the authors could strengthen their work by extending their algorithm to additional cross-domain settings. This would provide further validation of the algorithm's effectiveness and generalizability.

Recommendation:
I recommend rejecting this paper in its current form, primarily due to the weaknesses mentioned above.

**Additional Comments On Reviewer Discussion:**

During the discussion phase, the authors provided a detailed discussion about the encoder-decoder identification issue and included additional experiments regarding label accuracy. While they partially addressed the concerns raised by i9LB, I believe a more comprehensive study on the preference labels is still necessary to fully address the identified gaps.

---

### Decision · Program_Chairs · 2025-01-22

Reject